# Efficacy of *Bifidobacterium animalis* subsp. *lactis* BL-99 in the treatment of functional dyspepsia: a randomized placebo-controlled clinical trial

Qi Zhang ®[1,12], Guang Li[2,12], Wen Zhao ®[1,12], Xifan Wang ®[3,12], Jingjing He ®[1,12], Limian Zhou ®[4], Xiaoxu Zhang ®[5], Peng An ®[1], Yinghua Liu ®[6], Chengying Zhang ®[7], Yong Zhang ®[6], Simin Liu[8], Liang Zhao ®[5], Rong Liu ®[1], Yixuan Li[1], Wenjian Jiang[9], Xiaoyu Wang ®[1], Qingyu Wang ®[10], Bing Fang ®[1], Yuyang Zhao ®[5], Yimei Ren ®[5], Xiaokang Niu ®[1], Dongjie Li ®[9], Shaoqi Shi ®[1], Wei-Lian Hung ®[11,13] ✉, Ran Wang ®[5,13] ✉, Xinjuan Liu ®[2,13] ✉ & Fazheng Ren ®[1,13] ✉

Current treatment for functional dyspepsia (FD) has limited and unsustainable efficacy. Probiotics have the sustainable potential to alleviate FD. This randomized controlled clinical trial (Chinese Clinical Trial Registry, ChiCTR2000041430) assigned 200 FD patients to receive placebo, positive-drug (rabeprazole), or *Bifidobacterium animalis* subsp. *lactis* BL-99 (BL-99; low, high doses) for 8-week. The primary outcome was the clinical response rate (CRR) of FD score after 8-week treatment. The secondary outcomes were CRR of FD score at other periods, and PDS, EPS, serum indicators, fecal microbiota and metabolites. The CRR in FD score for the BL-99_high group [45 (90.0%)] was significantly higher than that for placebo [29 (58.0%), $p = 0.001$], BL-99_low [37 (74.0%), $p = 0.044$] and positive_control [35 (70.0%), $p = 0.017$] groups after 8-week treatment. This effect was sustained until 2-week after treatment but disappeared 8-week after treatment. Further metagenomic and metabolomics revealed that BL-99 promoted the accumulation of SCFA-producing microbiota and the increase of SCFA levels in stool and serum, which may account for the increase of serum gastrin level. This study supports the potential use of BL-99 for the treatment of FD.

Functional dyspepsia (FD) is a common chronic gastrointestinal disorder without known organic lesions[1]. The global prevalence of uninvestigated dyspepsia is 21%[2]. Epidemiological investigations have revealed that the symptoms vary in approximately two-thirds of patients with FD irrespective of postprandial distress syndrome (PDS; with postprandial fullness and early satiety symptoms) or epigastric pain syndrome (EPS; with epigastric pain and epigastric burning symptoms) subtypes[3]. In patients with FD, these symptoms are persistent for at least 1–3 days per week and last for more than 3 months[4]. Recurrent and prolonged symptoms contribute to poor quality of life and high medical expenses[5]. Therefore, searching for prolonged treatment of FD has considerable clinical value.

---

The pathophysiological mechanisms of FD have not been elucidated. Previous studies have proposed several distinct mechanisms for FD, including gastroduodenal motor disorders, visceral hypersensitivity, brain-gut interactions, and subtle duodenal inflammation[6]. Various anti-dyspepsia drugs (acid-suppressive therapy, prokinetics, neuro-modulators, and herbal therapies) have been recommended to treat FD symptoms in clinical practice. However, these drugs are associated with side effects and unknown long-term efficacy[7]. Among them, proton pump inhibitors (PPIs) are considered to be the most effective first-line therapy for FD although their long-term efficacy is limited, which may be related to changes in fecal microbiota and the increased risk of intestinal infection[8]. Hence, it is an urgent need to develop an efficient, targeted, and long-term therapy for FD.

Probiotics, which exert beneficial effects on health, generally colonize the cecum and colon owing to the harsh conditions in the gastrointestinal tract[9,10]. Some studies have exerted probiotics' potent therapeutic effects on FD. *Lactobacillus gasseri* OLL2716, *Lactobacillus paracasei* LC-37, *Bacillus coagulans* MY01, and *B. subtilis* MY02 can significantly alleviate postprandial discomfort, epigastric pain, belching, and other FD symptoms[11–13]. Wauters et al. also demonstrated that the efficacy of probiotics on FD was associated with the abundance of *Faecalibacterium* in feces[13]. In addition, changes in serum pepsinogen and gastrin by administration of probiotics have already been reported[14–16]. However, the therapeutic effects of probiotics vary depending on the bacterial strain. *Bifidobacterium animalis* subsp. *lactis* BL-99 (BL-99, GenBank accession number: OP748915) was isolated from the feces of a healthy infant. In vitro and in vivo experiments showed that BL-99 was a non-pathogenic and safe strain[17]. BL-99 was reported to alleviate intestinal inflammation in mice with osteoporosis and colitis[18,19]. Besides, the ability of probiotics to colonize the body indicates that their beneficial effects may be sustained, which is promising for long-term relief of FD. So the long-term efficacy of BL-99 alleviating FD also needs to be further investigated.

Here, we show that the clinical response rate (CRR) of FD score for BL-99 is significantly higher after an 8-week treatment compared to the control group. This effect is sustained until 2-week after treatment but disappears 8 weeks after treatment. These results highlight the potential effect of BL-99 in the treatment of FD.

## Results

### Baseline characteristics

Between 26 December 2020 and 10 February 2021, 336 consecutive FD patients were screened and assessed for eligibility. A total of 123 individuals were excluded because they did not meet the inclusion criteria ($n = 104$), withdrew consent ($n = 12$), or had other reasons ($n = 7$). After enrollment, 13 patients were excluded due to withdrawal of consent ($n = 8$) or other reasons ($n = 5$), leaving a total of 200 patients who were then randomly assigned to four groups. Among these, 185 (92.5%) completed the entire clinical trial (45, 48, 47, and 45 in the placebo, positive_control, BL-99_low, and BL-99_high group, respectively; Fig. 1). Baseline characteristics are summarized in Table 1. The four treatment groups had similar baseline characteristics. The mean age of the participants was 51.43 years, the mean BMI was 25.24 kg/m², and 74.5% were female. The mean FD scores in the placebo, positive_control, BL-99_low, and BL-99_high groups were 1.60, 1.61, 1.62, and 1.88 at baseline, respectively.

### Effect of BL-99 treatment on FD symptoms

The results of the effect on CRR based on the intention-to-treat (ITT) set are presented in Table 2. The primary outcome, 8-week CRR of FD score was significantly higher for BL-99_high [45 (90.0%)] than placebo [29 (58.0%), $p = 0.001$], BL-99_low [37 (74.0%), $p = 0.044$] and positive_control group [35 (70.0%), $p = 0.017$]. At the 2-week follow-up after the treatment, the CRR of FD score in the BL-99_high group [42 (84.0%)] was still significantly higher than placebo [31 (62.0%),

$p = 0.016$] and positive_control group [33 (66.0%), $p = 0.041$], but there was no significant difference between the BL-99_high and BL-99_low groups. Post-treatment follow-up at 8 weeks no longer showed significant differences in CRR between the 4 groups. Similar results were observed in the per-protocol (PP) set, which are presented in Supplementary Table 1. It also showed that the high dose of BL-99 group [43 (95.6%)] had a significantly higher 8-week-treatment CRR of FD score compared to placebo [28 (62.2%), $p = 0.001$], BL-99_low [36 (76.6%), $p = 0.019$] and positive_control group [34 (70.8%), $p = 0.006$]. The results for post-treatment 2-week and post-treatment 8-week CRR of FD score in the PP analysis were also similar to that of the ITT analysis.

Regarding the EPS score, 8-week CRR in the BL-99_high group [37 (74.0%)] was significantly higher than placebo [24 (48.0%), $p = 0.009$] and positive drug group [27 (54.0%), $p = 0.039$]. Even 8 weeks after the treatment, the BL-99_high group still had significantly higher CRR than the placebo and BL-99_low group. The results of the PP analysis were consistent with this.

As for the PDS score, the 4-week CRR in the BL-99_high group was consistently higher than the placebo and positive drug group in both the ITT and PP analyses. This difference between groups persisted at the 2-week follow-up after the treatment only in the PP analysis, but not in the ITT analysis. No significant differences between the four groups were observed at other study points (visit or survey).

The results of CRR were analyzed separately in men and women, in patients with BMI < 24 and BMI ≥ 24 kg/m² based on ITT as well (Supplementary Tables 2–5). Overall, the results of the primary outcome for women, BMI < 24 and BMI ≥ 24 kg/m² were consistent with those for the total population. Due to the relatively small proportion of male participants (25.5%), no significant effects were found in men.

As a post hoc analysis, data on patients who became symptom-free after treatment were also presented based on the ITT set (Supplementary Table 6). It shows that the proportion of people with no FD symptoms was significantly higher in the BL-99_high group (39 [78.0%]) than that in the placebo (18 [36.0%], $p < 0.001$) and the positive_control (21 [42.0%], $p < 0.001$) groups at the 2-week follow-up after the treatment, but not at 8-week treatment. Similar results were observed for the proportion of those with no PDS symptom both at 8-week treatment and 2 weeks after the treatment.

In addition, The means of FD, PDS, and EPS scores were also compared among the four groups (Supplementary Tables 7 and 8). The results showed that there were no significant differences in FD, EPS, or PDS scores between the 4 treatment groups at any visit or survey in either the ITT or PP analyses (all $p_{overall}$ values are ≥0.05).

Moreover, treatment with probiotic and dyspepsia drugs was safe compared with placebo, with a similar incidence of all adverse events [2 (4.0%), 1 (2.0%), 1 (2.0%), and 1 (2.0%) of 50; Supplementary Table 9].

### BL-99 treatment changed the serum indicators in FD patients

The serum indicators were evaluated by measuring the serum pepsinogen I (PGI), pepsinogen II (PGII), pepsinogen ratio (PGR = PGI/PGII), and gastrin 17 (G17) of FD participants (Table 3). After 8 weeks of intervention, the increase in serum G17 from baseline in the BL-99_high group (mean = 4.11, SD = 4.73) and was significantly higher than that in the placebo group (mean = 0.14, SD = 1.60, $p < 0.001$), the positive_control group (mean = 0.78, SD = 3.06, $p < 0.001$) and the BL-99_low group (mean = 1.87, SD = 2.96, $p = 0.003$). At 2-week follow-up after the treatment, the changes in serum indicators in BL-99 groups were not significantly different from those in other treatment groups.

### BL-99 treatment remodeled the gut microbiota in FD patients

The fecal microbiome was comparatively analyzed before and after BL-99 intervention using high-throughput metagenomic shotgun sequencing. Principal coordinate analysis revealed that the gut microbial composition was similar between the four groups at baseline (permutational multivariate analysis of variance [PERMANOVA];

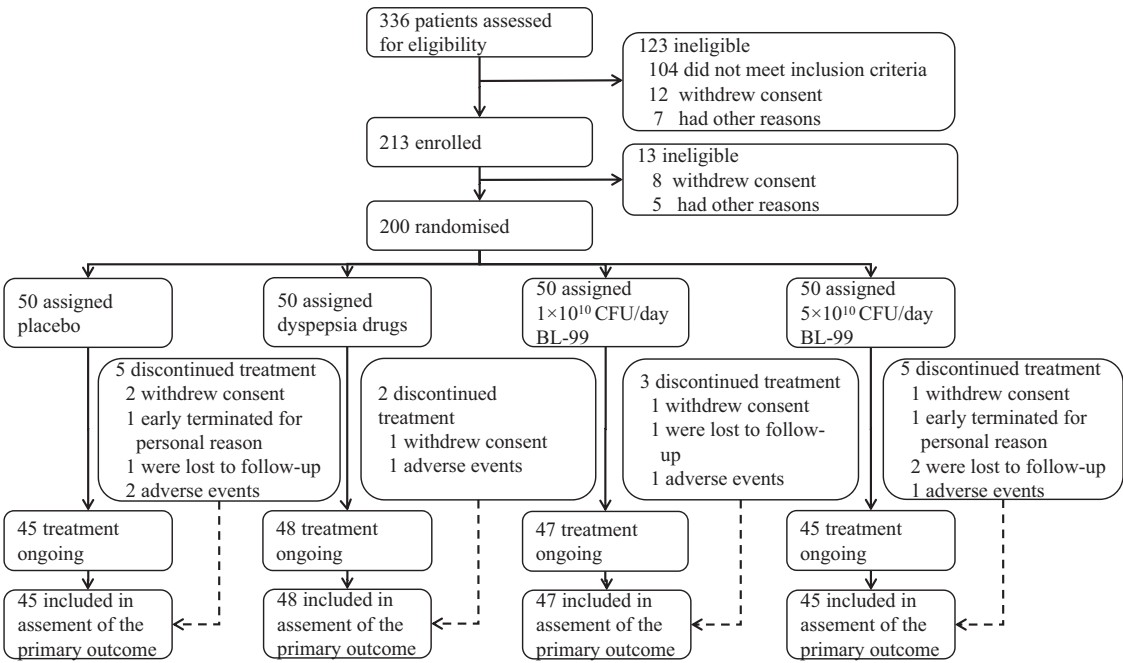

**Fig. 1 | Trial profile.** Other reasons include participant relocation, unable to be contacted, and time conflicts.

## Table 1 | Baseline characteristics of this study

| Characteristics | Placebo (n = 50) | Positive_control (n = 50) | BL-99_low (n = 50) | BL-99_high (n = 50) |
|---|---|---|---|---|
| Number of participants recruited from CCMU[a] | 30 | 32 | 30 | 31 |
| Number of participants recruited from CPLAGH[b] | 20 | 18 | 20 | 19 |
| Age, years | 50.12 (47.12–53.12) | 53.80 (52.05–55.55) | 51.60 (49.16–54.04) | 50.20 (47.00–53.40) |
| Gender[c], Male n (%): Female n (%) | 13(26.0%): 37 (74.0%) | 12(24.0%): 38 (76.0%) | 13(26.0%): 37 (74.0%) | 13(26.0%): 37 (74.0%) |
| BMI[d], kg/m² | 25.10 ± 3.78 | 24.73 ± 3.89 | 25.63 ± 4.42 | 25.48 ± 4.87 |
| Postprandial fullness score | 2.08 (1.85–2.31) | 1.98 (1.81–2.15) | 1.96 (1.70–2.22) | 2.24 (2.04–2.44) |
| Early satiety score | 1.76 (1.44–2.08) | 1.92 (1.75–2.09) | 1.94 (1.66–2.22) | 1.96 (1.69–2.23) |
| Epigastric pain score | 1.06 (0.71–1.41) | 1.08 (0.76–1.40) | 1.02 (0.68–1.36) | 1.36 (1.05–1.67) |
| Epigastric burning score | 1.50 (1.14–1.86) | 1.46 (1.14–1.78) | 1.56 (1.23–1.89) | 1.94 (1.65–2.23) |
| PDS score[e] | 1.92 (1.70–2.14) | 1.95 (1.82–2.08) | 1.95 (1.74–2.16) | 2.10 (1.91–2.30) |
| EPS score[f] | 1.28 (0.97–1.59) | 1.27 (1.00–1.54) | 1.29 (1.02–1.56) | 1.65 (1.41–1.89) |
| FD score[g] | 1.60 (1.35–1.85) | 1.61 (1.44–1.78) | 1.62 (1.41–1.83) | 1.88 (1.68–2.07) |

Data are presented as mean ± standard deviation, n (%), or mean (95% confidence interval). Patients in the placebo, positive_control, BL-99_low, and BL-99_high groups were administered with maltodextrin (2 g/day), rabeprazole (10 mg/ day), low-dose BL-99 (1×10¹⁰ CFU/day), and high-dose BL-99 (5×10¹⁰ CFU/day), respectively. Source data are provided as a Source Data file.
BL-99, *Bifidobacterium animalis* subsp. *lactis* BL-99.
[a]CCMU, Beijing Chaoyang Hospital, Capital Medical University.
[b]CPLAGH, Chinese PLA General Hospital.
[c]Gender data are expressed as a percentage of the population.
[d]BMI: body mass index (weight in kilograms divided by the square of the height in meters).
[e]PDS score: the postprandial distress syndrome score calculated as the mean of postprandial fullness score and early satiety score.
[f]EPS score: the epigastric pain syndrome score calculated as the mean of epigastric pain score and epigastric burning score.
[g]FD score: the composite functional dyspepsia score calculated as the mean of postprandial fullness, early satiety, epigastric pain, and epigastric burning scores.

$p = 0.121$). However, there was a significant change in the gut microbiota of subjects among different groups after the 8-week treatment (PERMANOVA $p = 0.037$; Fig. 2a and Supplementary Fig. 1). The most marked change was observed in the BL-99_high group, which exhibited a decreased abundance of *Bacteroidetes* and an increased abundance of *Firmicutes* after treatment relative to the baseline (Fig. 2b and Supplementary Fig. 2). At the species level, the average relative abundances of *B. animalis*, which exhibited the highest variation, in the BL-99_low and BL-99_high groups were 0.3% and 1.0%, respectively, after treatment. Additionally, BL-99 supplementation significantly increased the abundances of two short-chain fatty acid (SCFA)-producing bacteria (*Faecalibacterium prausnitzii* and *Roseburia*

*intestinalis*) and two lactate-producing *Ligilactobacillus* spp. (*L. ruminis* and *L. salivarius*) after intervention. In contrast, BL-99 supplementation decreased the abundances of some *Bacteroidetes* species, such as *Bacteroides uniformis*, *Bacteroides thetaiotaomicron*, *Phocaeicola vulgatus*, *Alistipes putredinis*, and *Alistipes shahii* (Fig. 2c and Supplementary Fig. 3). Functional analysis of the gut microbiome revealed that the abundance of SCFA synthetases in the BL-99_high group was significantly upregulated after intervention, while the decomposition of bile acids and toxins was markedly upregulated (Fig. 2d). These results indicate that the modulation of gut microbiota composition, especially the upregulation of SCFA-producing bacteria, maybe a potential mechanism through which BL-99 alleviates FD. The relative

**Table 2 | The clinical response rate for all (men and women) participants based on intention-to-treat (ITT) set**

| Clinical response rate[a], No. (%) | | Placebo (n = 50) | Positive_control (n = 50) | BL-99_low (n = 50) | BL-99_high (n = 50) | $p_{overall}$ | $p$ Positive_control vs. Placebo | BL-99_low vs. Placebo | BL-99 high vs. Placebo | BL-99_low vs. Positive_control | BL-99_high vs. Positive_control | BL-99_high vs. BL-99_low |
|---|---|---|---|---|---|---|---|---|---|---|---|---|
| 4-week treatment | FD score[b] | 29 (58.0) | 28 (56.0) | 28 (56.0) | 38 (76.0) | 0.113 | – | – | – | – | – | – |
| | PDS score[c] | 31 (62.0) | 32 (64.0) | 37 (74.0) | 43 (86.0) | 0.031 | 0.440 | 0.200 | 0.008 | 0.281 | 0.014 | 0.139 |
| | EPS score[d] | 24 (48.0) | 23 (46.0) | 25 (50.0) | 35 (70.0) | 0.059 | – | – | – | – | – | – |
| 8-week treatment | FD score | 29 (58.0) | 35 (70.0) | 37 (74.0) | 45 (90.0) | 0.004 | 0.213 | 0.094 | 0.001 | 0.656 | 0.017 | 0.044 |
| | PDS score | 34 (68.0) | 39 (78.0) | 40 80.0) | 44 (88.0) | 0.111 | – | – | – | – | – | – |
| | EPS score | 24 (48.0) | 27 (54.0) | 31 (62.0) | 37 (74.0) | 0.049 | 0.549 | 0.161 | 0.009 | 0.418 | 0.039 | 0.200 |
| 2-week follow-up | FD score | 31 (62.0) | 33 (66.0) | 38 (76.0) | 42 (84.0) | 0.049 | 0.173 | 0.133 | 0.016 | 0.272 | 0.041 | 0.320 |
| | PDS score | 36 (72.0) | 38 (76.0) | 44 (88.0) | 44 (88.0) | 0.085 | – | – | – | – | – | – |
| | EPS score | 26 (52.0) | 30 (60.0) | 32 (64.0) | 38 (76.0) | 0.092 | – | – | – | – | – | – |
| 8-week questionnaire survey | FD score | 10 (20.0) | 8 (16.0) | 9 (18.0) | 16 (32.0) | 0.204 | – | – | – | – | – | – |
| | PDS score | 12 (24.0) | 16 (32.0) | 18 (36.0) | 19 (38.0) | 0.454 | – | – | – | – | – | – |
| | EPS score | 5 (10.0) | 9 (18.0) | 6 (12.0) | 15 (30.0) | 0.038 | 0.255 | 0.750 | 0.017 | 0.404 | 0.164 | 0.032 |

All hypothesis tests were two-sided. $p < 0.05$ was considered significant. Patients in the placebo, positive_control, BL-99_low, and BL-99_high groups were administered with maltodextrin (2 g/day), rabeprazole (10 mg/day), low-dose BL-99 (1 × 10¹⁰ CFU/day), and high-dose BL-99 (5 × 10¹⁰ CFU/day), respectively. Source data are provided as a Source Data file.
BL-99, *Bifidobacterium animalis* subsp. *lactis* BL-99.

[a]Clinical response rate was defined as the proportion of participants with a score (i.e., FD score, PDS score, and EPS score) decrease >0.5.

[b]FD score: the composite functional dyspepsia score is calculated as the mean of postprandial fullness, early satiety, epigastric pain, and epigastric burning scores.

[c]PDS score: the postprandial distress syndrome score is calculated as the mean of postprandial fullness score and early satiety score.

[d]EPS score: the epigastric pain syndrome score is calculated as the mean of epigastric pain score and epigastric burning score.

**Table 3 | Within-group changes in functional dyspepsia–relative serum indexes**

| Change in serum indexes/ (ng/ml) | | Placebo (n=45) | Positive_control (n=48) | BL-99_low (n=47) | BL-99_high (n=45) | p_overall | p Positive_control vs. Placebo | BL-99_low vs. Placebo | BL-99_high vs. Placebo | BL-99_low vs. Positive_control | BL-99_high vs. Positive_control | BL-99_high vs. BL-99_low |
|---|---|---|---|---|---|---|---|---|---|---|---|---|
| From the baseline to the post-treatment period | PGI[a] | 3.39±19.65 | 14.00±26.37 | 2.79±30.253 | 10.58±20.70 | 0.109 | 0.053 | 0.917 | 0.207 | 0.039 | 0.525 | 0.169 |
| | PGII[b] | −0.45±6.29 | −2.38±11.04 | −0.61±3.69 | −2.24±5.86 | − | − | − | − | − | − | − |
| | PGR[c] | 0.85±4.52 | 4.59±6.25 | 0.72±2.87 | 2.23±6.95 | 0.003 | 0.001 | 0.916 | 0.256 | 0.001 | 0.046 | 0.221 |
| | G17[d] | 0.14±1.60 | 0.78±3.06 | 1.87±2.96 | 4.11±4.73 | <0.001 | 0.378 | 0.024 | <0.001 | 0.130 | <0.001 | 0.003 |
| From baseline to the 2-week follow-up period | PGI | −2.82±14.48 | −3.02±19.82 | −1.31±15.00 | −2.15±12.65 | − | − | − | − | − | − | − |
| | PGII | −1.26±5.84 | −3.44±3.60 | −1.18±6.06 | −0.64±5.56 | 0.060 | 0.060 | 0.942 | 0.600 | 0.060 | 0.013 | 0.647 |
| | PGR | 0.18±4.45 | 2.68±4.32 | 0.77±3.65 | −0.54±4.05 | 0.003 | 0.007 | 0.543 | 0.447 | 0.036 | <0.001 | 0.168 |
| | G17 | −0.47±8.19 | 0.11±3.66 | 0.35±4.29 | 0.33±6.09 | − | − | − | − | − | − | − |

Data are presented as mean ± standard deviation (SD) in the per-protocol (PP) set. Patients in the placebo, positive_control, BL-99_low, and BL-99_high groups were administered with maltodextrin (2 g/day), rabeprazole (10 mg/day), low-dose BL-99 ($5 \times 10^{10}$ CFU/day), and high-dose BL-99 ($1 \times 10^{10}$ CFU/day), respectively. All hypothesis tests were two-sided. One-way analysis of variance with least significant difference method was used to analyze the differences among the four groups, and $p < 0.05$ was considered significant. Source data are provided as a Source Data file.
BL-99, *Bifidobacterium animalis* subsp. *lactis* BL-99.
[a]PGI: pepsinogen I.
[b]PGII: pepsinogen II.
[c]PGR: pepsinogen ratio = PGI/PGII.
[d]G17: Gastrin 17.

abundance of *Bifidobacterium animalis* was quantified at 2-week follow-up after treatment. The *Bifidobacterium animalis* abundance in the BL-99_low and BL-99_high groups was higher than that in the positive_control group (Fig. 2e), suggesting the prolonged efficacy of BL-99.

### BL-99 treatment increased fecal and serum SCFA levels

The fecal metabolome of subjects was profiled using untargeted metabolomics (Supplementary Methods). Similar to the gut microbiome profiles, the fecal metabolome profiles of the four groups were not significantly different at the baseline (PERMANOVA; $p = 0.419$), but were significantly different after 8 weeks of treatment (PERMANOVA; $p = 0.008$; Supplementary Fig. 4). The fecal abundances of three SCFAs (acetate, propanoate, and butyrate) were upregulated in the BL-99_high group after intervention (Fig. 3a). However, the abundances of acetate, propanoate, and butyrate were not significantly different in the other three groups. Targeted analysis of fecal acetate, propanoate, and butyrate confirmed the upregulation of butyrate in the BL-99_high group after intervention (Fig. 3b). The serum acetate, propanoate, and butyrate concentrations were significantly upregulated in the positive_control, BL-99_low, and BL-99_high groups after intervention (Fig. 3c). Random forest models were used to assess the effect size of gut microbiota on the fecal and serum concentrations of SCFAs. Gut microbiota markedly affected the fecal and serum concentrations of acetate, propanoate, and butyrate, contributing to an average of 12.7% (ranging from 3.3 to 23.0%) of variations in concentration (Fig. 3d). Some species, such as *Faecalibacterium prausnitzii* and *Ligilactobacillus ruminis* upregulated by BL-99 markedly affected the serum or fecal butyrate concentrations, while *Bifidobacterium animalis* markedly affected the fecal acetate and butyrate concentrations (Fig. 3e).

### Effect of SCFAs on serum gastrin

Acetate and butyrate were infused into the carotid artery of SD rats for 45 min to elucidate the effect of SCFA on serum gastrin. The serum gastrin levels were determined at 0, 15, 30, and 45 min respectively, and the detailed animal experiment method was shown in the Supplementary Methods. SCFA infusion results showed that 8 μmol/(kg-min) acetate, 20 μmol/(kg-min) acetate, and 1 μmol/(kg-min) butyrate increased serum gastrin levels after perfusion. Moreover, serum gastrin in the 20 μmol/(kg-min) acetate group ($365.68 \pm 25.03$ pg/ml) was significantly higher than that in the 8 μmol/(kg-min) acetate group ($270.27 \pm 23.00$ pg/ml, $p < 0.001$) and the 2 μmol/(kg-min) acetate group ($201.86 \pm 35.15$ pg/ml, $p = 0.001$) after acetate infusion for 45 min. Similar results were found in the butyrate infusion experiment, which showed that serum gastrin in the 1 μmol/(kg-min) acetate group ($310.10 \pm 25.94$ pg/ml) was significantly higher than that in the 0.5 μmol/(kg-min) acetate group ($210.50 \pm 26.68$ pg/ml, $p < 0.001$) and the 0.1 μmol/(kg-min) acetate group ($191.15 \pm 29.75$ pg/ml, $p < 0.001$) (Supplementary Fig. 5). This suggests that acetate and butyrate maybe affect the serum gastrin level of FD patients.

## Discussion

This randomized, parallel-group, positive-drug, and placebo-controlled clinical trial showed that BL-99 exhibited good efficacy in the treatment of functional dyspepsia. The CRR of FD score in the BL-99 group was significantly higher than that for placebo and positive drug after an 8-week treatment. This effect lasted up to 2 weeks after the treatment ended, but disappeared 8 weeks after treatment. Serological data revealed that BL-99 increased G17 levels after 8-week treatment. Fecal microbiome and metabolome analyses found that high-dose BL-99 treatment increased the abundance of SCFA-producing microbiota, SCFA-synthase, and SCFA content in serum and feces. Further, SCFA infusion experiments confirmed the correlation between SCFAs and gastrin. Therefore, BL-99 may affect the production of gastrin by improving the composition of gut microbiota and the production of SCFA, thus alleviating the FD symptoms.

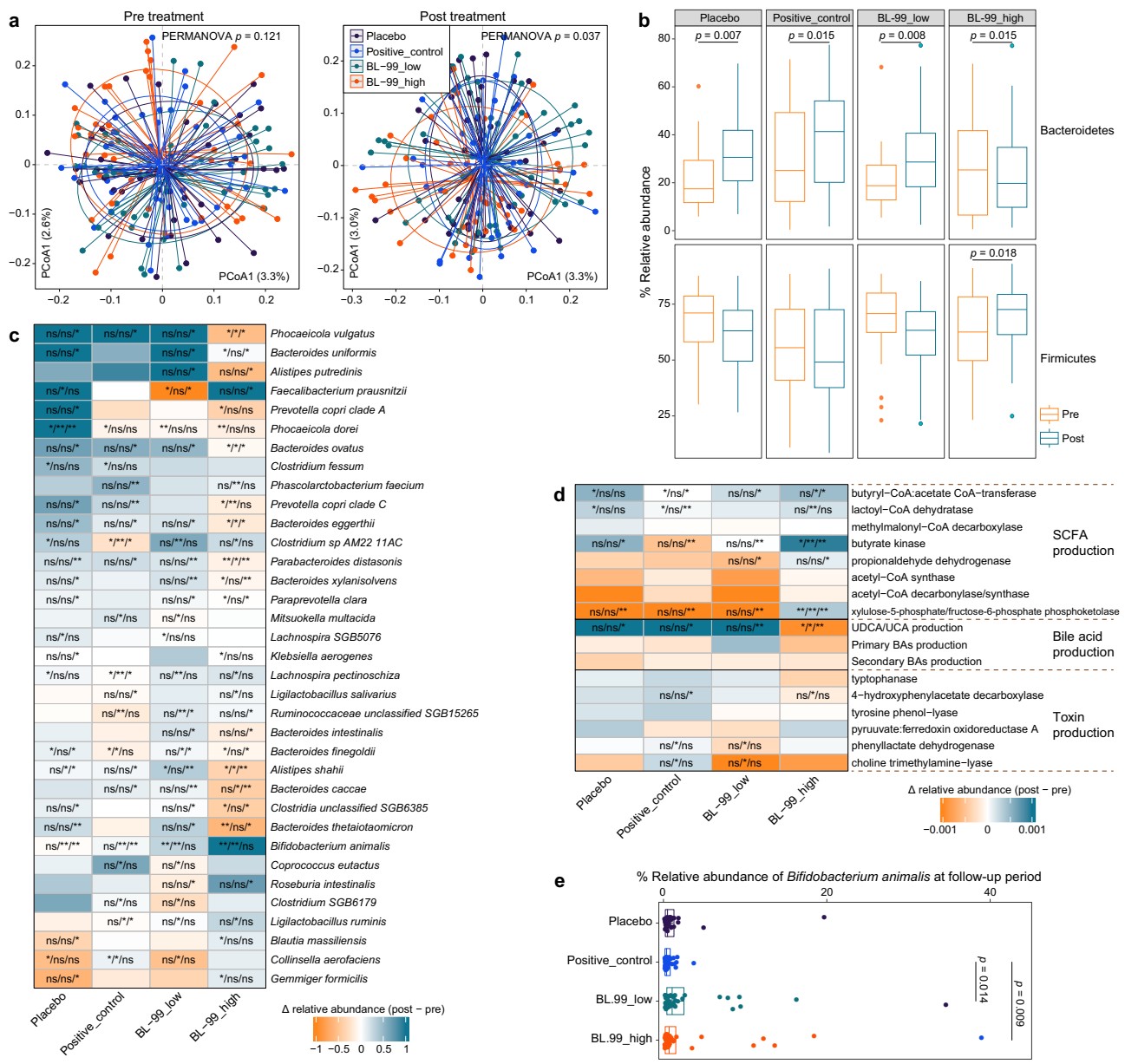

**Fig. 2 | Comparative analysis of the gut microbial composition in fecal samples from patients with functional dyspepsia (FD) treated with BL-99. a** Principal coordinate analysis (PCoA) of microbiota communities in the fecal samples among four groups at baseline and post-treatment period. Samples are shown at the first and second principal coordinates (PCoA1 and PCoA2), and the ratio of variance contributed by these two PCoAs is shown. Ellipsoids represent a 95% confidence interval surrounding each group. **b** Boxplot showing the relative abundances of *Bacteroidetes* and *Firmicutes* in samples at baseline and post-treatment period. Boxes represent the interquartile range between the first and third quartiles and the median (internal line). Whiskers denote the lowest and highest values within 1.5 times the range of the first and third quartiles, respectively. Dots represent outlier samples beyond the whiskers. *p* values are calculated using the two-side Wilcoxon rank-sum test. **c** Changes in the abundance of species from the baseline to the post-treatment period. Heatmap shows the changes in the mean relative abundance of species from the baseline to the post-treatment period in samples within each

group. For each species in each group, the significance levels of the comparisons between the changes in one group relative to the other three groups are calculated using the two-side Wilcoxon rank-sum test and denoted as follows: ns non-significant; *$p < 0.05$; **$p < 0.01$ (non-significant data in all comparisons are omitted). **d** Changes in microbial functions from the baseline to the post-treatment period. **e** The relative abundance of *Bifidobacterium animalis* among the four groups at the follow-up period. Boxes represent the interquartile range between the first and third quartiles and the median (internal line). Whiskers denote the lowest and highest values within 1.5 times the range of the first and third quartiles, respectively. Dots represent outlier samples beyond the whiskers. *p* values are calculated using the two-side Wilcoxon rank-sum test. Patients in the placebo ($n = 45$), positive_control ($n = 48$), BL-99_low ($n = 47$), and BL-99_high ($n = 45$) groups were administered with maltodextrin (2 g/day); rabeprazole (10 mg/day); low-dose BL-99 ($1 \times 10^{10}$ CFU/day), and high-dose BL-99 ($5 \times 10^{10}$ CFU/day), respectively. Source data are provided as a Source Data file.

When evaluating the treatment effect against FD symptoms, the selection of a primary outcome is crucial because FD patients have no obvious organic abnormalities and clear disease biomarkers[20]. Patients with FD defined by Rome Revision IV (2016) diagnostic criteria suffer from EPS or PDS symptoms; therefore, it is more understandable to choose symptom score as the primary outcome. The CRR of FD score

after 8-week treatment was chosen as the primary outcome in our study, which was based on a study conducted in University Hospitals Leuven that also evaluated the efficacy of probiotics in improving FD[13]. At the same time, a previous FD questionnaire validation study confirmed that a change of 0.5 was the threshold for the minimal clinically significant difference[13]. In addition, referring to similar studies[13,21], we

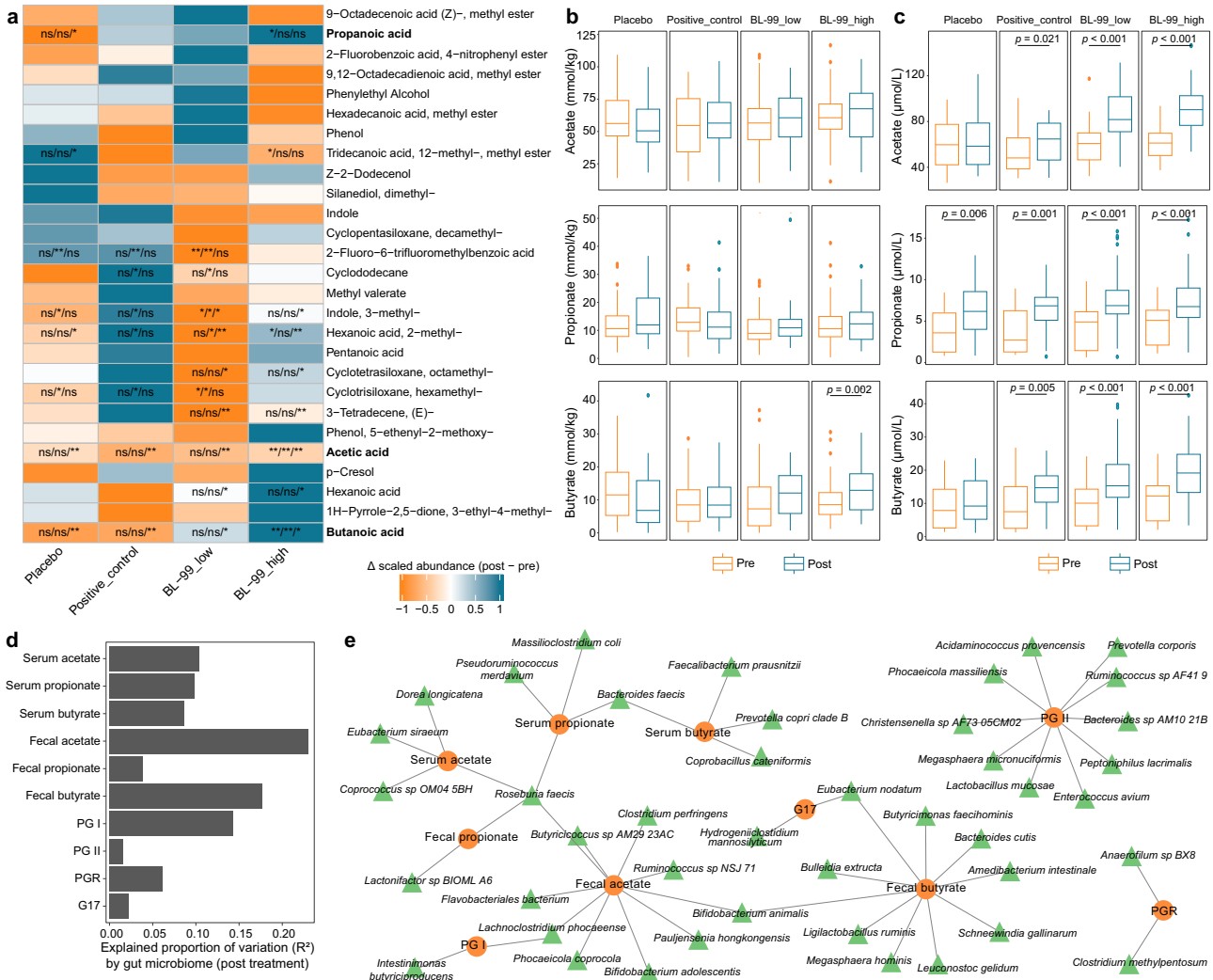

**Fig. 3 | Effects of BL-99 treatment on the fecal and serum metabolites in patients with functional dyspepsia. a** Changes in fecal metabolites from the baseline to the post-treatment period. Heatmap shows the changes in the mean relative abundance of metabolites from the baseline to the post-treatment period in samples within each group. For each species in each group, the significance levels of the comparisons between the changes in one group relative to the other three groups are calculated using the two-side Wilcoxon rank-sum test and denoted as follows: ns non-significant; *$p < 0.05$; **$p < 0.01$ (non-significant data in all comparisons are omitted). Boxplot showing the concentrations of short-chain fatty acids (SCFAs) in the fecal (**b**) and serum (**c**) samples at baseline and post-treatment period. Boxes represent the interquartile range between the first and third quartiles and the median (internal line). Whiskers denote the lowest and highest values within 1.5 times the range of the first and third quartiles, respectively. Dots represent outlier samples beyond the whiskers. $p$ values are calculated using the two-side Wilcoxon rank-sum test. **d** The explained variations of fecal and serum SCFAs and clinical parameters by the gut microbiome at the post-treatment time point. Bar plots indicate the explained variation (effect size $R^2$) of each metabolite or parameter. **e** Network view of gut species, fecal and serum SCFAs, and clinical parameters. Circles represent the SCFAs or clinical parameters, while the surrounding connected triangles represent the gut species that had the highest contributions to the SCFAs or parameters and were used in the random forest models. Patients in the placebo ($n = 45$), positive_control ($n = 48$), BL-99_low ($n = 47$), and BL-99_high ($n = 45$) groups were administered with maltodextrin (2 g/day), rabeprazole (10 mg/day), low-dose BL-99 ($1 \times 10^{10}$ CFU/day), and high-dose BL-99 ($5 \times 10^{10}$ CFU/day), respectively. Source data are provided as a Source Data file.

combined PDS and EPS score as FD score, which represented a comprehensive evaluation of FD symptoms.

Although PPIs are an internationally recommended medication for FD, their prolonged efficacy is limited and their side effects on intestinal health are irreversible. Meanwhile, PPI intake alters gut microbiota composition and increases the risk of intestinal infection, which suggests that safety and prolonged alternative therapies are needed[8]. Probiotics, colonize the intestine for a long time, and have potential advantages for the prolonged treatment of FD. And probiotics have been recommended as monotherapy or add-on therapy to treat FD symptoms. Ohtsu et al. indicated that the overall effect of *Lactobacillus gasseri* OLL2716 on gastric symptoms was more positive in *H. pylori*-negative FD individuals[22]. Wauters et al. showed similar efficacy in that a combination of *Bacillus coagulans* MY01 and *Bacillus*

*subtilis* MY02 was efficacious in relieving PDS or EPS symptoms[13]. Our results also demonstrated that the clinical response rate was significantly higher in the BL-99 group (90.0%) than in the placebo (58.0%), positive_control (70.0%) and BL-99_low (74.0%) groups after 8-week treatment. However, this effect only lasted up to 2 weeks after the intervention ended, but disappeared 8 weeks after treatment discontinuation.

Studies have confirmed that serum pepsinogen and gastrin levels in FD patients are different from those in healthy persons and are associated with various symptoms of FD[23–25]. For instance, Tahara et al.[24] discovered that serum PGII levels were significantly elevated and the PGI/II ratio was significantly reduced in both *H. pylori* positive and negative FD patients compared to healthy controls. Furthermore, Igarashi et al. found that *Lactobacillus gasseri* OLL2716 increased

serum PGI levels in FD patients and other functional upper gastro-intestinal disorders patients[26]. Additionally, G17, an important gastro-intestinal hormone, has also been reported to be potentially related to FD, and it was revealed that acupuncture, a type of traditional Chinese medical therapy, improved FD symptoms and increased serum G17 levels in FD patients[27]. Therefore, PGI, PGII, and G17 were determined in this study. Our results showed that BL-99 had no significant effect on serum PG level after 8-week treatment, but increased serum G17 levels in FD participants compared with placebo and positive_control groups. So our results indicated that BL-99 treatment could regulate the changes in gastrin associated with FD.

Although the pathogenesis of FD has not been elucidated, Wauters et al. demonstrated that *B. coagulans* MY01 and *B. subtilis* MY02 exerted potent therapeutic effects on FD by modulating the abundance of fecal *Faecalibacterium*[13]. Based on previous studies, this study examined the correlation between probiotics, fecal microbiota and metabolites, and FD. The results of this study suggested that BL-99 intervention alleviated the symptoms of FD, altered the fecal micro-biota composition, and upregulated the abundance of SCFA-producing microbiota. Random forest models and contribution ana-lysis of fecal microbiota and metabolites revealed that the alleviation of FD symptoms was dependent on the abundance of *Bifidobacterium animalis*, *Faecalibacterium prausnitzii*, and *Roseburia intestinali*. In summary, BL-99 alleviated the symptoms of FD, altered the composi-tion of fecal microbiota, and upregulated the abundance of SCFA-producing microbiota.

Moreover, SCFA has been reported to affect FD symptoms by improving gastrointestinal motility and intestinal epithelial barrier function. Sun et al. showed that *Lactobacillus paracasei* LC-37 pro-moted an increase in fecal acetic acid, propionic acid, and butyric acid content in FD patients[12]. Not only did increased fecal SCFA levels detected, but also increased serum SCFA contents were also found in our exploratory study. At the same time, butyrate, an important energy substance, could alleviate gastrointestinal mucosal atrophy. The increase in the butyrate-producing genus *Roseburia*[28] also confirmed that BL-99 was better for the improvement of FD symptoms compared with the placebo or positive drug. Moreover, the primary activity of organic acids is related to the decrease in gastric pH, which could help to convert inactive pepsinogen into active pepsin for efficient proteolysis[29]. Meanwhile, studies have confirmed that SCFA, produced by gut microbiota metabolism can stimulate parasympathetic nerve activation, thereby stimulating gastric G cells to secrete gastrin[30–32]. To demonstrate the correlation between SCFAs and serum index, SCFAs were infused into SD rats, and the results showed that acetate and butyrate could stimulate serum gastrin level. These results showed that BL-99 changed the gut microbiota communication, and promoted the increase of SCFA in feces and serum, further promoting the increase of G17, thus alleviating FD symptoms.

This study had some limitations. Firstly, as this is a hospital-based study that recruited patients from outpatient clinics, the results may not be generalizable to the general FD population, such as those in the community. Secondly, as only Chinese patients were recruited at the Beijing Chaoyang Hospital, Capital Medical University (CCMU; Beijing, China) and Chinese PLA General Hospital (CPLAGH; Beijing, China), the effectiveness of BL-99 in patients with FD from different countries, ethnicities, and clinical backgrounds was not evaluated. Thirdly, con-sidering the participants' wishes, we did not perform an endoscopy to collect gastroduodenal biopsies for mucosal-associated microbiome (MAM) detection. However, as probiotics mainly colonize the cecum and colon, we hypothesized that it exerts health effects primarily by regulating gut microbiota, which was also confirmed by our results. Fourthly, although FD participants were required to maintain their dietary habits during the study, no dietary survey was conducted to assess the effects of diet on gut microbiota composition. Fifthly, it should be noted that this study exclusively focuses on the efficacy of a specific strain (BL-99) in improving FD symptoms. Therefore, caution must be exercised when extrapolating these findings to other strains, as further research is required to investigate the effects of alternative strains. Sixthly, the limited representation of male participants (25.5%) in this study may have resulted in inadequate statistical power and thus invalidated the primary outcome among men. Consequently, this study does not provide conclusive evidence regarding gender differ-ences, which should be further investigated with larger sample sizes. Seventhly, the positive-drug group was not blinded in this study, so the participants' subjectivity may affect the accuracy of symptom report-ing, which may bias the results. However, since the double-blind method was successfully implemented in the BL-99_high, BL-99_low, and placebo groups, the effect of BL-99 relative to placebo should be credible, and the results of these 3 groups also confirmed this. In addition, even if the positive-drug group was not blinded, the clinical response rate of the BL-99_high group was still higher than that of the drug group after the 8-week treatment, which further supports the conclusions of this study.

Nevertheless, the strengths of this study include the rigorous study design with sufficient clinical data, which were provided by serology and multi-omics studies during the treatment and follow-up periods. Professional physicians evaluated the FD participants based on strict inclusion and exclusion criteria, reducing the influence of other potential factors on the participants' source. The probiotic 16S rRNA sequencing results showed that BL-99 possessed potential adhesion genes[33], which provided the possibility for the prolonged efficacy of BL-99 in patients with FD (Supplementary Fig. 6 and Table 10). Previous studies evaluating the therapeutic effects of pro-biotics on FD have focused on the alleviation of FD symptoms. How-ever, this study used a multi-omics approach to analyze the effect of BL-99 on the composition and function of gut microbiota, as well as on the FD-related metabolites. Moreover, SCFA infusion experiments demonstrated that metabolites of gut microbiota can affect serum gastrin level in mice, suggesting that the observed increase of serum gastrin in BL-99_high group could be related to the accumulation of SCFA-promoting gut microbiota.

In conclusion, *Bifidobacterium* BL-99 showed good efficacy in patients with FD. The higher CRR after an 8-week treatment and 2-week follow-up period, and remarkably higher *Bifidobacterium animalis* relative abundance further confirmed the efficacy of BL-99. And this effect may be related to SCFA-producing microbiota, serum and fecal SCFA, and serum G17. This study highlights the potential role of pro-biotics in FD, which are informative for the design of larger multi-center, multi-ethnic groups, and multi-subtype trials.

## Methods

### Study design and participants

A randomized, parallel-group, positive-drug, and placebo-controlled clinical trial was performed in Beijing, China. The study was approved by the Institutional Review Board of CCMU (No.2020-ke-497) and was performed by the Declaration of Helsinki. The study was registered at Chictr.org.cn with a registration number of ChiCTR2000041430.

Outpatients (18–60 years) with FD symptoms were recruited and screened between 26 December 2020 and 10 February 2021 at CCMU and CPLAGH. Inclusion criteria included meeting the diagnostic cri-teria for FD of Rome IV[34]. All FD patients had normal upper endoscopy results within 1 year before enrollment. Patients with any symptoms of acute diarrhea, gastroesophageal reflux disease, irritable bowel syn-drome (IBS), defecation problems, severe systemic (cardiovascular, liver, kidney, or hematopoietic) diseases, *Helicobacter pylori* infection (diagnosed by the $C^{14}$-urea breath test), or use of immunosuppressant drugs, antibiotics in the past 3 months were excluded. And patients treated with FD-related medications within 6 months before the study were excluded. All participants provided written informed consent before inclusion, and were compensated fairly in accordance with the

requirements of the Institutional Review Board of CCMU, without any inducements.

## Randomization and blinding

We used computer-generated random numbers to establish simple randomized grouping sequences. Eligible participants were identified by clinicians, and information was then transmitted via telephone, or email to a specialized statistician who had no further role in the trial to determine the treatment allocation based on the pre-established allocation sequence, which was concealed until all participants were allocated. Participants were randomly assigned (1: 1: 1: 1) to 4 groups, which included the placebo, positive_control (only PPI treatment), low-dose probiotic, and high-dose probiotic groups (only BL-99 treatment). Due to the difficulty of making probiotic formulations identical to PPI drugs, blinding was not possible in all four treatment groups. The positive-drug group was treated with PPI drugs, and the other three groups received solid beverage powder with identical appearance, taste, and smell between groups. Therefore, the positive-drug group was open-label. For the other three groups, researchers and participants were blinded to treatment assignments until the study was completed.

## Study procedures

The study procedures are shown in Supplementary Fig. 7.

All included participants first underwent a 2-week run-in period. During the run-in period, participants were not allowed to take foods containing probiotics (such as probiotic powder, probiotic yogurt, etc.). Then participants were treated with PPI, BL-99, or placebo for 8 weeks, followed by an 8-week post-treatment follow-up. During the treatment, participants were instructed to maintain their habitual lifestyle habits such as diet and physical activity and were not allowed to take antibiotics. A total of 4 visits [at baseline (V1), 4-week treatment (V2), 8-week treatment (V3), and 2-week follow-up (V4)] and 1 survey [questionnaire surveys 8 weeks after the treatment (V5)] were conducted throughout the study period. At each visit and the final survey, participants were surveyed using a uniform FD symptom assessment questionnaire (see the "Symptom assessment" section for details). Blood and fecal samples were collected at V1, V3, and V4. Even 8 weeks post the treatment, we were fortunate to receive the questionnaire responses from all participants who completed the treatment.

## Treatment

All eligible participants received one of the following four treatments: placebo: 2 g/day maltodextrin (batch number: 2020122401); positive_control: 10 mg/day rabeprazole (one kind of PPI, batch number: 1711033); low dose probiotic: 2 g/day solid beverage containing $1 \times 10^{10}$ CFU/day BL-99 (batch number: 2020122402); and high dose probiotic: 2 g solid beverage containing $5 \times 10^{10}$ CFU/day (batch number: 2020122403). All the treatments were performed once daily. The maltodextrin and BL-99 were manufactured by Beijing Heyiyuan Biotechnology Co., Ltd (Beijing, China). Treatment compliance was determined by counting the empty solid beverage bars returned by the participants, which was defined as good if several empty bars accounted for 80% or more of the total number sent out.

## Symptom assessment

FD symptoms were assessed by a previously validated questionnaire[35–37], which is shown in Supplementary Note 1. Specifically, FD symptoms included postprandial fullness, early satiety, epigastric pain, and epigastric burning, with a score range of 0-3 for each symptom (0 = none, 1 = mild, 2 = moderate, and 3 = severe). FD score was calculated as the average of the four symptoms. PDS score was the average score of postprandial fullness and early satiety, and the EPS score was the average score of the remaining two symptoms. In this study, the symptoms of each participant were assessed by two

professional gastroenterologists, and the final FD symptom score was determined after consultation.

## Serum indicators measurement

Blood samples were collected via venipuncture after the participants have fasted overnight. Serum was then extracted to measure gastrin17 (G17), pepsinogen I (PGI), and pepsinogen II (PGII) using a Biohit enzyme-linked immunosorbent assay (ELISA) kit (Biohit, Oyj, Finland). The pepsinogen ratio (PGR) is the ratio of PGI to PGII. Serum short-chain fatty acids (SCFA) were also detected using the Agilent GC-8860 gas chromatograph instrument with FFAP column (30 m * 250 μm * 0.25 μm, Agilent Technologies, Inc., USA)[38], more detailed methods were shown in Supplementary Methods.

## Fecal sample collection and metagenome and metabolome detection

Fresh fecal samples were collected and placed in sterile retention bottles. Then the stool samples were immediately placed on ice, transported to the laboratory within 1 h, and frozen at −80 °C for subsequent use[39]. More importantly, fecal samples were homogenized by Bertin Precellys Evolution sample homogenizer (Bertin Technologies SAS, France)[40,41], and then the homogenized fecal samples were randomly weighed for further index detection. Fecal metagenomics was measured by the Whole-metagenome shotgun sequencing based on the lumina NovaSeq PE150 (Illumina Inc., San Diego, CA, USA) platform at Majorbio Bio-Pharm Technology Co., Ltd. (Shanghai, China). And fecal metabolite features were analyzed by Agilent GC-Q-TOF-7200-7890B with DB-WAX capillary column (30 m × 250 μm × 0.25 μm, Agilent Technologies, Inc., USA)[42], more detailed methods were shown in Supplementary Methods.

## Outcomes

The primary outcome was the clinical response rate (CRR) of the FD score at week 8 of treatment. Clinical response was defined as a score (i.e., FD score, PDS score, and EPS score) decrease >0.5. CRR was then calculated as the proportion of clinical responders[13]. The secondary endpoints were CRR of FD score at week 4 of treatment, week 2 and 8 of follow-up; CRR of PDS score and EPS score at every visit or survey after initiation of treatment; changes in serum indicators (PGI, PGII, PGR, and G17), fecal microbiota, fecal metabolites, and changes of SCFA in feces and serum from baseline to 8-week treatment and 2-week follow-up periods.

Subjects were asked to report any adverse effects during the treatment and follow-up periods, such as bloating, nausea, diarrhea, itchy skin, etc. Safety was assessed by classifying adverse events using the Common Terminology Criteria for Adverse Events (CTCAE) version 5.0 at each study period or in the case of early termination.

## Statistical analysis

In a study of probiotics improving FD, the CRRs after 8 weeks of treatment of probiotic ($5 \times 10^9$ CFU /day) and placebo were 48% and 20%, respectively[13], which were thus assumed for the low-dose probiotic group ($1 \times 10^{10}$ CFU/day, 48%) and the placebo group (20%) in our sample size calculation. In addition, we assumed a CRR of 50% for the positive drug group based on a study[43], and an intermediate value of 49% for the high-dose probiotic group, which was between the low-dose probiotic group and the positive drug group. Based on these assumptions, a sample size of 42 would be required per group (power of 80% and two-sided $\alpha = 0.05$). Considering a 20% dropout rate, 50 subjects would be needed to be included in each group (200 for 4 groups).

The main data set for efficacy analysis in this study was the ITT set, which included all participants who were randomized. In the ITT set, missing values for symptom scores were imputed based on the last observation carried forward method. Violations that significantly

affect efficacy included (but were not limited to) the following: (1) Received interference therapy after inclusion; (2) Poor compliance (e.g., with follow-up visits less than 80% of the required number of visits); (3) Follow-up beyond the window period[44]. We also analyzed symptom scores based on the PP set, which referred to participants who had completed the planned treatment and visits according to the protocol and had no obvious effect on the therapeutic effect.

Gender in this study was determined based on self-report. Analyses were conducted separately in the total population, in men, women, BMI ≤ 24 and BMI > 24 kg/m² based on ITT. Continuous variables were described as mean and standard deviation (SD). Counting variables were described as frequency and percentage. For comparison between groups, one-way analysis of variance (ANOVA) or Kruskal–Wallis rank test was used for continuous variables, and the chi-square test was used for counting variables. If the overall difference between the groups was significant, the least significant difference (LSD) method was used for multiple comparisons. All hypothesis tests were two-sided. $p < 0.05$ was considered significant. Statistical analyses were performed using SPSS Statistics version 24 (SPSS Institute, Chicago, IL, USA). Figures were drawn using GraphPad Prism 9.0.0 (GraphPad Software, The North Parker, USA).

For metagenomic and metabolomic analyses, principal coordinates analysis (PCoA) and distance-based redundancy analysis (dbRDA) were performed based on the Bray-Curtis dissimilarity on the gut microbial composition using *capscale* function (R *vegan* 2.6.4 package). Permutational multivariate analysis of variance (PERMANOVA, effect size analysis) was performed with the adonis function of the R *vegan* 2.6.4 package, and the *adonis p* value was generated based on 1000 permutations.

### Reporting summary

Further information on research design is available in the Nature Portfolio Reporting Summary linked to this article.

## Data availability

The demographic characteristics, symptoms of functional dyspepsia, and serum markers data generated in this study are provided in the Supplementary Information/Source Data file. The Metagenomic sequencing data generated in this study have been deposited in the NCBI Sequence Read Archive (SRA) database under the accession code PRJNA936638. Mass spectral raw data generated in this study have been deposited in the MetaboLights database under the accession code MTBLS7169. The clinical study protocol and statistical analysis plan file are provided in Supplementary Note 2 and 3, respectively. The other data supporting the findings of this study are available within the paper and additional files. Source data are provided with this paper.

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

## Acknowledgements

Metagenomics analysis was performed using the online platform of Majorbio Cloud Platform (www.majorbio.com). The spelling, grammar, sentence structure, and terminology of our manuscript were edited by Elsevier Language Editing Services and Bullet Edits Limited. This work was financially supported by the National Key R&D Program of China (grant number 2021YFD1600204, F.R.), The 111 project from the Education Ministry of China (No. B18053, R.W.), the Key project of multidisciplinary Clinical Research Innovation Team of Beijing Chaoyang Hospital Affiliated to Capital Medical University (CYDXK202207, X.L.), and the 2020 Science and Technology for Mongolia Program (2020-KJXM-GCZX-2, W.L.H.).

## Author contributions

F.R., X.L., R.W., W.L.H., Q.Z., G.L., and W.Z. formulated the research questions and developed the study protocol. Q.Z., G.L., W.Z., X.F.W., J.H., L.M.Z., and X.Z. were responsible for patient recruitment and obtaining informed consent signatures. J.H., P.A., Y.L., C.Y.Z., Y.Z., and S.L. were responsible for the random allocation of patients. Q.Z., L.Z., Y.L., W.J., Y.Y.Z., and Y.R. collected and extracted the data, with X.N., D.L., and S.S., providing supervision. Q.Z., W.Z., X.F.W., J.H., P.A., and R.W. analyzed the data. Q.Z., G.L., W.Z., X.F.W., J.H., S.L., Y.L., and W.J. wrote the manuscript. F.R., X.L., R.L., R.W., W.L.H., X.Y.W., Q.W., and B.F. revised the manuscript. All authors had full access to all the data in the study, reviewed the manuscript, and had final responsibility for the decision to submit for publication.

## Competing interests

The authors declare no competing interests.

## Additional information

[1]Key Laboratory of Functional Dairy, Co-constructed by Ministry of Education and Beijing Government, Department of Nutrition and Health, China Agricultural University, Beijing, China. [2]Department of Gastroenterology, Beijing Chaoyang Hospital, Capital Medical University, Beijing, China. [3]Department of Obstetrics and Gynecology, Columbia University, New York, NY, USA. [4]School of Food and Biological Engineering, Hefei University of Technology, Hefei, China. [5]Beijing Advanced Innovation Center for Food Nutrition and Human Health, China Agricultural University, Beijing, China. [6]Department of Nutrition, Chinese PLA

General Hospital, Beijing, China. [7]Department of General Practice, The Third Medical Center of Chinese PLA General Hospital, Beijing, China. [8]Center for Global Cardiometabolic Health, Departments of Epidemiology, Medicine, and Surgery, Brown University, Providence, RI, USA. [9]Department of Cardiovascular Surgery, Beijing Anzhen Hospital, Capital Medical University, Beijing, China. [10]Academy of Medical Sciences, Beijing Hospital/National Center of Gerontology of National Health Commission, Beijing, China. [11]National Center of Technology Innovation for Dairy, Inner Mongolia Dairy Technology Research Institute Co. Ltd., Hohhot, China. [12]These authors contributed equally: Qi Zhang, Guang Li, Wen Zhao, Xifan Wang, Jingjing He. [13]These authors jointly supervised this work: Wei-Lian Hung, Ran Wang, Xinjuan Liu, Fazheng Ren. ✉e-mail: hongweilian@yili.com; wangran@cau.edu.cn; Liuxinjuan@mail.ccmu.edu.cn; renfazheng@263.net

