## [Peer Review File · Nature Communications]

REVIEWER EXPERTISE

Reviewer #1. Gastroenterology, functional dyspepsia, microbiome, clinical trials.

Reviewer #2. Probiotics, functional dyspepsia.

Reviewer #3. Clinical trial statistics.

REVIEWER COMMENTS

Reviewer #1 (Remarks to the Author):

Functional dyspepsia (FD) is a very common and unexplained GI disorder, and key area of unmet need because of its high prevalence (~10%), limited pharmaceutical options and major impact on quality of life. Recent evidence strongly suggests the duodenal microbiome is abnormal in FD, and low grade duodenal inflammation has been observed associated with intestinal permeability change and immune activation. This has led to a focus on the small intestine in FD rather than the stomach (in the absence of *H. pylori*). There is little evidence the colon or colonic microbiome plays a major role in the pathophysiology of FD.

In this randomised controlled trial, the authors investigated if a probiotic (*B. lactis*-99) in high or lower dose and a PPI improved FD symptoms, and assessed if stool microbiome (shotgun) and metabolomic profile was associated with response. The study was adequately powered.

Specific comments

The methods need to be clearer. When I first read this I thought it was a factorial design but I think you mean based on the flow chart and protocol only one arm received PPI and the probiotic arms did NOT receive PPI. Is this correct?

The primary outcome is specified as a decrease of greater than 0.5 out of 3 for total clinical symptom score from baseline to 8 weeks. However, assessment of symptoms is unclear. Was this by a validated questionnaire? Or clinical interview (and if so how was this standardised)? Exactly what individual symptoms were measured and graded?

Importantly, why was this particular primary outcome applied? Has it been used in other major trials? Is the change clinically meaningful? What was the distribution of individual and total symptom scores rather than "response rates" (please graph)?

The placebo response was high (>60%). This may reflect measurement bias. How many had symptom resolution (no symptoms) on therapy in each group? If very few, how many had only very minimal symptoms in each group?

How was concealed allocation assured?

Why was an intention to treat analysis not provided?

Are patients included in this trial reflective of FD in outpatient practice? Are the results generalisable?

What PPI was used? All the same drug? Why 10mg per day (low dose?)? How many not on PPI had previously been taking their drug and for how long (as it's a standard therapy for FD)? What about neuromodulators and other drugs in the study population by arm?

When were stool samples obtained exactly? Stored how? This should be clear in the main methods.

How many enrolled met Rome IV criteria? Bloating is NOT an FD symptom according to Rome IV - postprandial fullness is an FD symptom. Was this measured? Bloating could mean distention or just a feeling of gas!

Did this FD cohort undergo endoscopy to confirm the diagnosis pre-trial? Do you have gastroduodenal biopsies?

How many with FD in this study also had IBS, a well described overlap syndrome that is associated with a more severe phenotype? Did this influence outcome?

Was anxiety/depression assessed, a common comorbidity? Did this improve with probiotic therapy?

Why were these probiotic doses chosen? Was there preliminary data using these doses? How was probiotic efficacy established in terms of number of live organisms present before ingestion?

Why the focus on changes in pepsinogen and gastrin? These are at best very indirect assessment of gastric mucosal function. By the way this has nothing to do with gastric "digestion" (line 109). Plus where did you anticipate the probiotic would be acting? Does it act in the stomach at all? Or is the action more in the small intestinal microbiome where recent research has focussed in FD?

The shotgun metagenomic work is competently presented but the figure panels are hard to read. The faecal and serum metabolite data are of interest (fig. 5). Why not use this to guide the functional analyses rather than pulling out what seems at random multiple database results and presenting them in fig. 4?

The stool microbiome shotgun studies and metabolomics however may not reflect proximal actions of the probiotic, or changes at the mucosal associated microbiome (MAM) anywhere along the GI tract. The lack of upper GI biopsies with MAM results is a major limitation in interpreting this work.

Reviewer #2 (Remarks to the Author):

In this manuscript (NCOMMS-23-04238), the authors reported prolonged efficacy of *B. lactis* in the treatment of functional dyspepsia (FD), and concluded that restoration of dysbiotic microbiota in the gut is an underlying mechanism of the alleviation of symptoms in FD patients treated with a probiotic strain.

Comments:

1. Although significant link of gut microbiota has been considered to be involved in the pathophysiology of functional GI disorders (FGID), almost of them have been about irritable bowel syndrome (IBS), not about FD (eg, Wauters et al. *Gut* 2020;69:591). With regard to the microbiota in the pathophysiology underlying FD, small intestinal bacterial overgrowth (SIBO) and gastroduodenal dysbiosis appear to attract much attention in the investigation of FD so far.

However, in the authors' study, gut microbiome and their bacterial metabolites were exclusively examined, but no analysis about microbiomes of stomach, duodenum or proximal small intestine was done; no consideration about them was found even in Discussion. Thus, the scope of the authors' study seems so deviated. In order to stress the importance of gut microbiota in the pathophysiology of FD, authors cited Reference No. 21 (Gut 2018;67:778) and mentioned in the text that "there was significant changes of gut microbiota composition in FD patients". But this referred article described about IBS patients but not mentioned about FD patients at all. Nevertheless, it would have been very interesting to know the authors' opinion how do a wrong change in the gut microbiota and their metabolites distort the function of stomach and duodenum in FD, which are anatomically away from distal small and large intestines in which a lot of gut microbiota colonize.

2. In this study, a significant beneficial clinical response was obtained in the probiotic-treated group when compared with the placebo-treated group. In addition, probiotic treatment in FD patients converted dysbiotic microbiome into those colonizing the gut of healthy subject. Although there is a coincidence of those events, it exists little evidence indicating causal relationship between them. While the authors indicated a significant correlation between SCFA/serum index and gut microbiome (Fig. 6), it will make little sense to demonstrate evidence that gut microbiota is really involved in probiotic-induced improvement FD symptoms in their study.

3. Although the authors measured serum PGI, PGII and G17 before and after probiotic treatment and presented the within-group change in FD patients (Table 2). However, there was no data indicating some significant difference in the values of those parameters between FD patients and healthy control subjects without probiotic treatment in their study. Without such fundamental evidence or some supporting citations, why were the authors able to imply involvement of those serum activities in the alleviation of FD symptoms.

A change in serum PGI and PGII by administration of probiotics are already reported (Aliment Pharmacol Ther 2006;23:1077).

4. There are some wrong citations. For example, the authors mentioned in detail (Lines 214-218) that "Some scholars compared,,,24 FD patients and 21 age-matched,,,in the control group." Although that citation appears very important to support the authors' opinion, no article was referred for that.

The reason why Reference No. 21 is inappropriate is already shown in Comment 1.

Reviewer #3 (Remarks to the Author):

There is some useful information about the impact of probiotics for this condition in this population.

However, there is a lack of clarity around some of the methods and data analysis, especially with regards the primary outcome of the clinical trial which means I would not recommend publication. There is no acknowledgement of the disadvantages of a per-protocol analysis, nor the subjectivity of the outcome given no blinding in the trial.

The methods section needs much more detail as outlined below, as there is currently not enough detail to replicate the methods or to aid understanding of the impact of the intervention on clinical symptoms.

Abstract: The primary outcome should be identified in the abstract in the methods section. Also it should be outlined how the clinical response rates were measured, so the results section needs to be reduced to ensure this information can be included. In the results section I would also suggest indicating that there was no evidence for a difference between the BL-99 high group and BL-99 low group after the 8 week follow up period.

In general, for transparency - it should be written somewhere in the manuscript who sponsored and funded this clinical trial. I would also suggest the clinical trials registration information goes in the abstract but that may not be common in this journal. As a minor point - there were also a couple of typos and one unfinished sentence.

Introduction: There is an incorrect citation. 'Global prevalence estimate' comes from the following paper (AC Ford, A Marwaha, R Sood, P Moayyedi. Global prevalence of, and risk factors for, uninvestigated dyspepsia: a meta-analysis. *Gut*, 64 (2015), pp. 1049-1057) not reference 2 which simply refers to the paper mentioned above.

Results: It is important to list the 'other' reasons for exclusion, in particular for those that were enrolled but excluded prior to randomisation. More information should be given about what was done at the beginning of the 2-week run in period and what was done at the start of treatment. Were the same questionnaires used at every visit? This is not clear.

It is difficult to understand the 'average decrease' value as a clinical response rate. The average decrease should be an absolute number per participant and so doesn't link well into the clinical response rate which is a percentage of the group having a decrease above a certain level. I am not sure but I think if you are saying what proportion in each group had a decrease in their EPS+PDS score of >0.5 then that is not the 'average'. Elsewhere in the publication the word 'average' is not used which I think is better. I think it would also be important to present the mean (sd) of the scores individually (i.e PDS and EPS) and combined (PDS+EPS) at 8 weeks. Also the combination of PDS+EPS scores should be presented in the baseline table (Table 1).

Also in Table 1 it would be good to understand how the Total score was calculated. It does not appear to be the addition of clinical symptoms for all the symptoms.

Baseline characteristics across groups should not be compared as the randomisation is designed to give similar distributions so the footnote regarding this should be removed from Table 1.

In the legend to the table S1 and S2 and Figure 2 it should be outlined what significant level each letter corresponds without needing to refer back to the main text. I also think that the values for Figure 2 should be reported in a table, not just the different values for male and females as is currently in the appendix.

Importantly it is not clear why the numbers in each group in Table S1 for the primary outcome do not match those included in the analysis in Figure 1. There is mention of a per-protocol analysis but no definition in the text of how per-protocol was defined. It is not sufficient to have this just in the SAP and it should be outlined in the methods section.

Discussion:

There needs to be acknowledgement of the subjectivity of patient reported clinical symptoms and using symptom scores. This was not a blinded trial so all participants had knowledge of their allocation, or if it was blinded then there are not details of the blinding in this manuscript. Thus, assuming no blinding, those getting the strongest intervention potentially knew they were getting it and may report lower clinical symptom scores.

There also needs to be discussion of the limitation of per protocol analyses and the fact that this introduces bias that randomisation is trying to remove.

There is no justification of the choice of endpoint and why scores from PDS and EPS were combined and thus that needs to be added to the discussion section.

Methods:

It should be made clear it was run at two hospital sites and not one (as the first sentences infers) and potentially give recruitment numbers per group per site in Table 1.

As a minor point the word 'human' is not necessary before the word trial. That is made clear from the beginning.

As mentioned in the results section the definition of clinical response rate needs to be linked to the fact that it is reported as a percentage. So it needs to have a denominator and numerator so if it is the number that have the decrease >0.5 then it should be written as such.

Under the sample size section it should be made more explicit what information was used from reference 16 to input into the sample size calculation. It should be clear if the comparison was made on the percentages across the arms and if so what percentages were the assumptions based on (was the difference hypothesised to be the same for each group compared to placebo?). Also, there is no consideration of multiple testing and that three arms are being tested against the placebo.

It needs to be made clear that there was not a visit at the end of 8 weeks after treatment finished and it was only a survey. Were the completeness rates different and if so this should be made clear.

There should be a statement about missing data in the main text of the methods section. There is a line referenced in the STORMS/STROBE checklist, it did not seem to be about missing data particularly for primary and secondary outcomes.

Point-by-point response to the reviewers' comments

Reviewer #1 (Remarks to the Author):

Functional dyspepsia (FD) is a very common and unexplained GI disorder, and key area of unmet need because of its high prevalence (~10%), limited pharmaceutical options and major impact on quality of life. Recent evidence strongly suggests the duodenal microbiome is abnormal in FD, and low grade duodenal inflammation has been observed associated with intestinal permeability change and immune activation. This has led to a focus on the small intestine in FD rather than the stomach (in the absence of *H. pylori*). There is little evidence the colon or colonic microbiome plays a major role in the pathophysiology of FD.

In this randomised controlled trial, the authors investigated if a probiotic (*B. lactis*-99) in high or lower dose and a PPI improved FD symptoms, and assessed if stool microbiome (shotgun) and metabolomic profile was associated with response. The study was adequately powered.

Response: Thank you very much for your overall positive comments and constructive suggestions. Our detailed responses are as follows.

Specific comments

Question 1: The methods need to be clearer. When I first read this I thought it was a factorial design but I think you mean based on the flow chart and protocol only one arm received PPI and the probiotic arms did NOT receive PPI. Is this correct?

Response: We thank the reviewer for the valuable input. As suggested, we now provide more details of the method which makes it clearer. Our study was not a factorial design, but a randomized, parallel-group, positive-drug, and placebo-controlled clinical trial. We recruited 200 FD patients and randomized them into four groups, including: placebo group, positive control group (only PPI treatment), low- and high-dose *Bifidobacterium animalis* BL-99 groups (only BL-99 treatment). As the reviewer suggested, we have modified our description in the revised methods, and made it more precise. We write now (Page 10, lines 302-304): 'Participants were randomly assigned (1: 1: 1: 1) to 4 groups, which included the placebo, positive control (only PPI treatment), low-dose probiotic, and high-dose probiotic groups (only BL-99 treatment).'

Question 2: The primary outcome is specified as a decrease of greater than 0.5 out of 3 for total clinical symptom score from baseline to 8 weeks. However, assessment of symptoms is unclear. Was this by a validated questionnaire? Or clinical interview (and if so how was this standardised)? Exactly what individual symptoms were measured and graded?

Response: We thank the reviewer for this valuable comment. The FD symptoms were assessed by previously reported FD symptom questionnaire¹⁻³, which was shown in the **Supplementary Note 1**. The questionnaire was evaluated by two professional gastroenterologists, and the FD symptom score was obtained after consultation. Specifically, FD symptoms included postprandial fullness, early satiety, epigastric pain, and epigastric burning, with a score range of 0-3 for each symptom. The method section was revised on Page 11, lines 337-344: 'FD symptoms were assessed by a previously validated questionnaire, which is shown in **Supplementary Note 1**. Specifically, FD symptoms included postprandial fullness, early satiety, epigastric pain, and

epigastric burning, with a score range of 0-3 for each symptom (0 = none, 1 = mild, 2 = moderate, and 3 = severe). FD score was calculated as the average of the four symptoms. PDS score was the average score of postprandial fullness and early satiety, and the EPS score was the average score of the remaining two symptoms. In this study, the symptoms of each participant were assessed by two professional gastroenterologists, and the final FD symptom score was determined after consultation.'

Reference

1. Ghoshal, U. C. et al. Development, translation and validation of enhanced Asian Rome III questionnaires for diagnosis of functional bowel diseases in major Asian languages: A Rome foundation-Asian neuro gastroenterology and motility association working team report. *J. Neurogastroenterol. Motil.* **21**, 83-92 (2015).
2. Tack, J. et al. Symptoms associated with hypersensitivity to gastric distention in functional dyspepsia. *Gastroenterology (New York, N.Y. 1943).* **121**, 526-535 (2001).
3. Cheong, P. K. et al. Low-dose imipramine for refractory functional dyspepsia: a randomised, double-blind, placebo-controlled trial. *Lancet Gastroenterol. Hepatol.* **3**, 837-844 (2018).

Question 3: Importantly, why was this particular primary outcome applied? Has it been used in other major trials? Is the change clinically meaningful? What was the distribution of individual and total symptom scores rather than "response rates" (please graph)?

Response: Thank you for your question. The clinical response rate of the FD score has been commonly used as primary outcome in FD related clinical studies. For example, it was used as the primary outcome to evaluate the efficacy of probiotics on FD¹. At the same time, a previous FD questionnaire validation study confirmed that a change of 0.5 was the threshold for the minimal clinically significant difference¹.

As the reviewer suggested, we graphed the distribution of individual and total symptom scores in the following **Fig. R1**. The FD score result showed that the number of people with no symptoms was higher in the BL-99_high group (28/62.2%) than that in the placebo group (21/46.7%) or the positive_control group (17/35.4%) after 8-week treatment. There were no significant differences between groups with minimal symptoms, and the number of people with moderate and severe symptoms was lower in the BL-99_high group (13/28.9%) than that in the placebo (20/44.4%) and positive_control group (22/45.8%).

Fig. R1. The distribution of individual and total symptom scores at 8-week treatment period.

a FD score: the composite functional dyspepsia score calculated as the mean of postprandial fullness, early satiety, epigastric pain, and epigastric burning scores. **b** PDS score: the postprandial distress syndrome score calculated as the mean of postprandial fullness score and early satiety score. **c** EPS score: the epigastric pain syndrome score calculated as the mean of epigastric pain score and epigastric burning score. **No symptoms:** Patients who had symptom resolution (no symptoms) after

8-week treatment; **Minimal symptoms:** Patients who have no more than two symptoms scores ≥ 1 after 8-week treatment; **Moderate and severe symptoms:** Patients except those who have no symptoms and those with minimal symptoms.

Reference

1. Wauters, L. et al. Efficacy and safety of spore-forming probiotics in the treatment of functional dyspepsia: a pilot randomised, double-blind, placebo-controlled trial. *Lancet Gastroenterol. Hepatol.* 6, 784-792 (2021).

Question 4: The placebo response was high (>60%). This may reflect measurement bias. How many had symptom resolution (no symptoms) on therapy in each group? If very few, how many had only very minimal symptoms in each group?

Response: Thank you for your comment. The occurrence of FD is related to the patient's psychological factors^{1,2}, which may trigger placebo effects, resulting in a high response rate in the placebo group³. Similarly, previous studies explored the effect of itopride on FD symptoms using a placebo as a control, with a response rate of 63% in the placebo group⁴. Based on your suggestions, we conducted a stratified analysis of symptom scores, and the results were shown in **Table R1**. The stratified analysis found that more people (28/62.2%) were no symptoms in the BL-99_high group than in the placebo group (21/46.7%) or the positive _control group (17/35.4%) after 8-week treatment; There were no significant differences between groups with minimal symptoms; The number of people with moderate and severe symptoms was lower in the BL-99_high group (13/28.9%) than in the placebo (20/44.4%) and positive _control group (22/45.8%). Therefore, although the response rate in the placebo group was high, BL-99 has a significant effect on FD symptom relief.

Table R1 Stratified analysis of symptom scores after 8-week treatment

Symptoms score		Placebo	Positive control	BL-99 low	BL-99 high
No./Total (%)		(n=45)	(n=48)	(n=47)	(n=45)
No symptoms	FD score	21 (46.7)	17 (35.4)	29 (61.7)	28 (62.2)
	PDS score	24 (53.3)	24 (50.0)	30 (63.8)	34 (75.6)
	EPS score	34 (75.6)	29 (60.4)	38 (80.9)	34 (75.6)
Minimal symptoms	FD score	4 (8.9)	3 (6.3)	3 (6.4)	4 (8.9)
	PDS score	3 (6.7)	2 (4.2)	3 (6.4)	2 (4.4)
	EPS score	1 (2.2)	4 (8.3)	2 (4.3)	2 (4.4)
Moderate and severe symptoms	FD score	20 (44.4)	28 (58.3)	15 (31.9)	13 (28.9)
	PDS score	18 (40.0)	22 (45.8)	14 (29.8)	9 (20.0)
	EPS score	10 (22.2)	15 (31.3)	7 (14.8)	9 (20.0)

Note: FD score: the composite functional dyspepsia score calculated as the mean of postprandial fullness, early satiety, epigastric pain, and epigastric burning scores. **PDS score:** the postprandial distress syndrome score calculated as the mean of postprandial fullness score and early satiety score. **EPS score:** the epigastric pain syndrome score calculated as the mean of the epigastric pain score and epigastric burning score. **No symptoms:** Patients who had symptom resolution (no symptoms) after 8-week treatment; **Minimal symptoms:** Patients who have no more than two symptoms scores ≥ 1 after 8-week treatment; **Moderate and severe symptoms:** Patients except those who have no symptoms and those with minimal symptoms.

Reference

1. Drossman, D. A. Functional gastrointestinal disorders: History, pathophysiology, clinical features and Rome

- IV. *Gastroenterology*, **16**, 00223-7 (2016).
2. Aro, P. et al. Anxiety is linked to new-onset dyspepsia in the Swedish population: A 10-year follow-up study. *Gastroenterology*. **148**, 928-937 (2015).
 3. Colloca, L. et al. Placebo and nocebo effects. *N. Engl. J. Med.* **382**, 554-561 (2020).
 4. Holtmann, G. et al. A placebo-controlled trial of itopride in functional dyspepsia. *N. Engl. J. Med.* **354**, 832-840 (2006).

Question 5: How was concealed allocation assured?

Response: We agree with the reviewer, and have provided detailed information on how concealed allocation was assured (Page 10, lines 298-302): 'We used computer-generated random numbers to establish simple randomized grouping sequences. Eligible participants were identified by clinicians, and information was then transmitted via telephone, or email to a specialized statistician who had no further role in the trial to determine the treatment allocation based on the pre-established allocation sequence, which was concealed until all participants were allocated.'

Question 6: Why was an intention to treat analysis not provided?

Response: Thank you for your question. In the revised manuscript, we analyzed symptom scores according to the intention-to-treat (ITT) principle (**Supplementary Table 1**). For the primary outcome, the ITT analysis yielded similar results to those of per-protocol (PP), which also supported our main conclusion (Pages 3-4, lines 80-86): 'Similar results were observed in the intention-to-treat (ITT) set, which are presented in **Supplementary Table 1**. It also showed that the high dose of BL-99 group [45 (90.0%)] had significantly higher 8-week-treatment CRR of FD score compared to placebo [29 (58.0%), $P = 0.001$], BL-99_low [37 (74.0%), $P = 0.044$] and positive control group [35 (70.0%), $P = 0.017$]. The results for post-treatment 2-week and post-treatment 8-week CRR of FD score in the ITT analysis were also similar to that of the PP analysis.'

Question 7: Are patients included in this trial reflective of FD in outpatient practice? Are the results generalisable?

Response: Thank you for your questions. FD patients are rarely hospitalized and outpatients were recruited. We have incorporated a more comprehensive description in the Methods section of revised manuscript (Page 10, lines 286-288): 'Outpatients (18-60 years) with FD symptoms were recruited and screened between 26 December 2020 and 10 February 2021 at Beijing Chaoyang Hospital, Capital Medical University (CCMU), and Chinese PLA General Hospital (CPLAGH).'

For the generalisation of our results, we have discussed in the Discussion section (Page 8, lines 244-246): 'Firstly, as this is a hospital-based study that recruited patients from outpatient clinics, the results may not be generalizable to the general FD population, such as those in the community.'

Reference

1. Hsu, Y. C. et al. Psychopathology and personality trait in subgroups of functional dyspepsia based on Rome III criteria. *Am. J. Gastroenterol.* **104**, 2534-2542 (2009).

Question 8: What PPI was used? All the same drug? Why 10mg per day (low dose)? How many not on PPI had previously been taking their drug and for how long (as it's a standard therapy for FD)? What about neuromodulators and other drugs in the study population by arm?

Response: We thank the reviewer for the helpful suggestion. In this study, the positive control group was intervened with the same PPI drug, namely rabeprazole. Based on previous studies and the Guidelines for Clinical Use of PPI, the standard dose of rabeprazole was 10 mg per day, not a low dose¹. We have added information about the PPI drug in the Methods section (Page 11, lines 327-332): 'All eligible participants received one of the following four treatments: placebo: 2 g/day maltodextrin (batch number: 2020122401); positive_control: 10 mg/day rabeprazole (one kind of PPI, batch number: 1711033); low dose probiotic: 2 g/day solid beverage containing 1×10^{10} CFU/day BL-99 (batch number: 2020122402); and high dose probiotic: 2 g solid beverage containing 5×10^{10} CFU/day (batch number: 2020122403). All the treatments were performed once daily.'

The FD patients recruited in this study had not received any FD-related drug treatment during the 6 months prior to the study (Page 10, lines 294-295): 'And patients treated with FD-related medications within 6 months before the study were excluded.'

In addition, we recorded the medication history of FD patients in the case report form (CRF) and did not find any individuals who used neuromodulators or other drugs.

Reference

1. Kamiya, T. et al. A multicenter randomized trial comparing rabeprazole and itopride in patients with functional dyspepsia in Japan: the NAGOYA study. *J. Clin. Biochem. Nutr.* **60**, 130-135 (2017).

Question 9: When were stool samples obtained exactly? Stored how? This should be clear in the main methods.

Response: Thank you for your suggestions. In this study, fresh fecal samples of each participant were collected in sterile retention bottles at baseline, 8-week treatment, and 2-week follow-up periods (Page 11, lines 322-323): 'Blood and fecal samples were collected at V1, V3, and V4.'

The stool samples were immediately placed on ice, transported to the laboratory within 1 h, and frozen at -80°C for subsequent use. The methods of the collection and storage of fecal samples were performed as described previously¹, and supplemented in the Methods section (Page 12, lines 353-355): 'Fresh fecal samples were collected and placed in sterile retention bottles. Then the stool samples were immediately placed on ice, transported to the laboratory within 1 h, and frozen at -80°C for subsequent use.'

Reference

1. Rossi, M. et al. Volatile organic compounds in feces associate with response to dietary intervention in patients with irritable bowel syndrome. *Clin. Gastroenterol. Hepatol.* **16**, 385-391 (2018).

Question 10: How many enrolled met Rome IV criteria? Bloating is NOT an FD symptom according to Rome IV - postprandial fullness is an FD symptom. Was this measured? Bloating could mean distention or just a feeling of gas!

Response: Thank you for your comments. In our study, all FD patients were strictly recruited by professional doctors according to the Rome IV diagnostic criteria. You are right that 'Bloating' was inappropriately used to express 'postprandial fullness'. Therefore, we have changed all 'bloating' to 'postprandial fullness' in the revised manuscript.

Question 11: Did this FD cohort undergo endoscopy to confirm the diagnosis pre-trial? Do you have

gastroduodenal biopsies?

Response: Thank you for your question. Participants were asked if they had undergone an endoscopic examination within one year prior to enrollment, and FD patients who had taken an examination and had normal upper gastrointestinal endoscopic results were included in this study (Page 10, lines 289-290): 'All FD patients had normal upper endoscopy results within one year before enrollment.'

Sorry that we don't have gastroduodenal biopsies. And we have added related information in the Discussion section (Page 9, lines 249-253): 'Thirdly, considering the participants' wishes, we did not perform an endoscopy to collect gastroduodenal biopsies for mucosal-associated microbiome (MAM) detection. However, as probiotics mainly colonize the cecum and colon, we hypothesized that it exerts health effects primarily by regulating gut microbiota, which was also confirmed by our results.'

Question 12: How many with FD in this study also had IBS, a well described overlap syndrome that is associated with a more severe phenotype? Did this influence outcome?

Response: Thank you for your questions. According to the Rome IV diagnostic criteria, the main difference between IBS and FD is whether patients have abdominal pain symptoms related to defecation or accompanied by changes in defecation habits¹. This study strictly recruited FD patients according to the Rome IV diagnostic criteria, with no IBS comorbidities. During recruitment, we inquired about the presence of comorbid conditions such as gastroesophageal reflux disease, IBS, or defecation problems, and excluded other diseases that may have overlapping symptoms with FD to avoid confounding effects on the primary outcome. We have also revised the Methods section (Page 10, lines 290-294): 'Patients with any symptoms of acute diarrhea, gastroesophageal reflux disease, irritable bowel syndrome (IBS), defecation problems, severe systemic (cardiovascular, liver, kidney, or hematopoietic) diseases, *Helicobacter pylori* infection (diagnosed by the C¹⁴-urea breath test), or use of immunosuppressant drugs, antibiotics in the past 3 months were excluded.'

Reference

1. Schmulson, M. J. et al. What is new in Rome IV. *J. Neurogastroenterol. Motil.* **23**, 151-163 (2017).

Question 13: Was anxiety/depression assessed, a common comorbidity? Did this improve with probiotic therapy?

Response: Thank you for your suggestion. Anxiety/depression is a comorbidity of FD, and there have been many studies on probiotics improving anxiety/depression¹, but this was not the focus of this study. The main purpose of this study was to investigate the prolonged efficacy of *Bifidobacterium* in the treatment of FD, but not a stratification analysis of anti-depression/anxiety drugs. In the future, we are interested in conducting new studies to explore the effects of probiotics on FD with comorbid depression/anxiety.

Reference

1. Tian, P. et al. *Bifidobacterium breve* CCFM1025 attenuates major depression disorder via regulating gut microbiome and tryptophan metabolism: A randomized clinical trial. *Brain, Behavior, and Immunity.* **100**, 233-241 (2022).

Question 14: Why were these probiotic doses chosen? Was there preliminary data using these doses? How was probiotic efficacy established in terms of number of live organisms present before

ingestion?

Response: Thank you for your questions. The International Scientific Association for Probiotics and Prebiotics (ISAPP) consensus and literature recommend that the common dose of *Bifidobacterium animalis* for improving gastrointestinal diseases is 1×10^{10} CFU/day^{1,2}. Our previous research results showed that *Bifidobacterium animalis* BL-99 could improve intestinal inflammation in mice with colitis at a dose of 1×10^9 CFU/d/mouse, a corresponding dose of human 1×10^{10} CFU/day according to a dosage conversion method³⁻⁵. Therefore, 1×10^{10} CFU/day was used as a low-dose treatment, and 5×10^{10} CFU/day was used as a high-dose treatment in this study.

Reference

1. Ibarra, A. et al. Effects of 28-day *Bifidobacterium animalis* subsp. *lactis* HN019 supplementation on colonic transit time and gastrointestinal symptoms in adults with functional constipation: A double-blind, randomized, placebo-controlled, and dose-ranging trial. *Gut Microbes*. **9**, 236-251 (2018).
2. Martoni, C. J. et al. *Lactobacillus acidophilus* DDS-1 and *Bifidobacterium lactis* UABla-12 improve abdominal pain severity and symptomology in irritable bowel syndrome: Randomized controlled trial. *Nutrients*. **12**, (2020).
3. Lan, H. et al. *Bifidobacterium lactis* BL-99 protects mice with osteoporosis caused by colitis via gut inflammation and gut microbiota regulation. *Food Funct*. **13**, 1482-1494 (2022).
4. Nan, X. et al. *Bifidobacterium animalis* subsp. *lactis* BL-99 ameliorates colitis-related lung injury in mice by modulating short-chain fatty acid production and inflammatory monocytes/macrophages. *Food Funct*. **14**, 1099-1112 (2023).
5. Nair, A. B. et al. A simple practice guide for dose conversion between animals and human. *J Basic Clin Pharm*. **7**, 27-31 (2016).

Question 15: Why the focus on changes in pepsinogen and gastrin? These are at best very indirect assessment of gastric mucosal function. By the way this has nothing to do with gastric "digestion" (line 109). Plus where did you anticipate the probiotic would be acting? Does it act in the stomach at all? Or is the action more in the small intestinal microbiome where recent research has focussed in FD?

Response: Thank you for your questions. Studies have confirmed that serum pepsinogen and gastrin levels in FD patients are different from those in healthy individuals and are associated with various symptoms of FD^{1,2}. The Maastricht V/Florence Consensus Report states that pepsinogen and gastrin are important parameters for managing FD³. Igarashi et al. found that *Lactobacillus gasseri* OLL2716 (LG21) increased serum PGI levels while alleviating FD symptoms⁴. Kwon et al. reported that different treatment methods increased serum G17 levels in FD patients⁵. Collectively, pepsinogen and gastrin can be used as objective indicators to characterize FD symptoms. Therefore, we focused on these two indicators. We have added related information in the Discussion section (Page 7, lines 204-209): 'Studies have confirmed that serum pepsinogen and gastrin levels in FD patients are different from those in healthy persons and are associated with various symptoms of FD. The Maastricht V/Florence Consensus Report states that pepsinogen and gastrin are important parameters for managing FD. Furthermore, Igarashi et al. found that *Lactobacillus gasseri* OLL2716 improved FD symptoms and altered serum PGI levels. Therefore, PGI, PGII, and G17 were determined in this study.'

Thank you for pointing out 'digestion'. We have made corrections to the relevant content.

We anticipate that probiotics work primarily through the lower digestive tract. This is mainly

because successful colonization of the gastrointestinal tract is a key factor for probiotics to be able to exert a sufficient host-interaction to confer health benefits⁶. At the same time, probiotics are generally more likely to colonize the cecum and colon due to the harsh conditions in the gastrointestinal tract^{7,8}. We found that after 8 weeks of BL-99 treatment, FD symptoms were significantly relieved, which were associated with promoting expansion of SCFA-producing microbiota of *Bifidobacterium animalis*, *Faecalibacterium prausnitzii*, and *Roseburia intestinalis*. Wauters, L. et al. also demonstrated that *B coagulans* MY01 and *B subtilis* MY02 were efficacious in the treatment of FD, and this efficacy was associated with the abundance of *Faecalibacterium* in feces⁹. These results confirm our anticipator.

Reference

1. Crafa, P. et al. Functional dyspepsia. *Acta Biomed.* **91**, e2020069 (2020).
2. Tahara, T. et al. Examination of serum pepsinogen in functional dyspepsia. *Hepatogastroenterology.* **59**, 2516-2522 (2012)
3. Malfertheiner, P. et al. Management of *Helicobacter pylori* infection—the Maastricht V/Florence Consensus Report. *Gut.* **66**, 6-30 (2016).
4. Igarashi, M. et al. Correlation between the serum pepsinogen I level and the symptom degree in proton pump inhibitor-users administered with a probiotic. *Pharmaceuticals.* **7**, 754-764 (2014).
5. Kwon, C. et al. Acupuncture as an add-on treatment for functional dyspepsia: A systematic review and meta-analysis. *Front. Med.* **8**, 682783 (2021).
6. Alp, D. et al. Adhesion mechanisms of lactic acid bacteria: conventional and novel approaches for testing. *World J. Microbiol. Biotechnol.* **35**, 156 (2019).
7. O'Hara, A. M. et al. The gut flora as a forgotten organ. *Embo Rep.* **7**, 688-693 (2006).
8. Taverniti, V. et al. Probiotics modulate mouse gut microbiota and influence intestinal immune and serotonergic gene expression in a site-specific fashion. *Front. Microbiol.* **12**, (2021).
9. Wauters, L. et al. Efficacy and safety of spore-forming probiotics in the treatment of functional dyspepsia: a pilot randomised, double-blind, placebo-controlled trial. *Lancet Gastroenterol. Hepatol.* **6**, 784-792 (2021)

Question 16: The shotgun metagenomic work is competently presented but the figure panels are hard to read. The faecal and serum metabolite data are of interest (fig. 5). Why not use this to guide the functional analyses rather than pulling out what seems at random multiple database results and presenting them in fig. 4?

Response: Thank you very much for your positive feedback and constructive suggestions. We have reorganized and redrawn the content about metagenomics and metabolomics. Based on the results of fecal and serum metabolites, we analyzed the abundance of SCFA synthetases and SCFA-producing microbiota. It was found that the abundance of SCFA synthetases and SCFA-producing microbiota in the BL-99_high group significantly increased after the 8-week treatment (**Fig. 2**). Random forest models were used to assess the effect size of gut microbiota on the fecal and serum concentrations of SCFAs. Gut microbiota markedly affected the fecal and serum concentrations of acetate, propanoate, and butyrate, contributing to an average of 12.7% (ranging from 3.3% to 23.0%) of variations in concentration (**Fig. 3d**). Some species, such as *Faecalibacterium prausnitzii* and *Ligilactobacillus ruminis* upregulated by BL-99 markedly affected the serum or fecal butyrate concentrations, while *Bifidobacterium animalis* markedly affected the fecal acetate and butyrate concentrations (**Fig. 3e**). It can be seen that the change of SCFA level of feces and serum is mainly caused by changes in the abundance of fecal microbiota.

Fig. 2. Comparative analysis of the gut microbial composition in fecal samples from patients with functional dyspepsia (FD) treated with BL-99. **a** Principal coordinate analysis (PCoA) of microbiota communities in the fecal samples among four groups at baseline and post-treatment period. Samples are shown at the first and second principal coordinates (PCoA1 and PCoA2), and the ratio of variance contributed by these two PCoAs is shown. Ellipsoids represent a 95% confidence interval surrounding each group. The below and left boxplots show the sample scores in PCoA1 and PCoA2 (boxes show medians/quartiles; error bars extend to the most extreme values within 1.5 interquartile ranges). **b** Boxplot showing the relative abundances of *Bacteroidetes* and *Firmicutes* in samples at baseline and post-treatment period. Boxes represent the interquartile range between the first and third quartiles and the median (internal line). Whiskers denote the lowest and highest values within 1.5 times the range of the first and third quartiles, respectively. Dots represent outlier samples beyond the whiskers. Wilcoxon rank-sum test. $*P < 0.05$; $**P < 0.01$; $***P < 0.001$. **c** Changes in the abundance of species from the baseline to the post-treatment period. Heatmap shows the changes in the mean relative abundance of species from the baseline to the post-treatment period in samples within each group. For each species in each group, the comparisons between the changes in one group relative to the other three groups are calculated using the Wilcoxon rank-sum test and denoted as follows: ns, non-significant; $*P < 0.05$; $**P < 0.01$ (non-significant data in all comparisons are omitted). **d** Changes in microbial functions from the baseline to the post-treatment period. **e** The relative abundance of *Bifidobacterium animalis* among the four groups at the follow-up period. Wilcoxon rank-sum test. $*P < 0.05$; $**P < 0.01$; $***P < 0.001$. Patients in the placebo, positive control, BL-99_low, and BL-99_low

groups were administered with maltodextrin (2 g/day); rabeprazole (10 mg/ day); low-dose BL-99 (1×10^{10} CFU/day), and high-dose BL-99 (5×10^{10} CFU/day), respectively.

Fig. 3. Effects of BL-99 treatment on the fecal and serum metabolites in patients with functional dyspepsia. **a** Changes in fecal metabolites from the baseline to the post-treatment period. Heatmap shows the changes in the mean relative abundance of metabolites from the baseline to the post-treatment period in samples within each group. For each species in each group, the significance levels of the comparisons between the changes in one group relative to the other three groups are calculated using the Wilcoxon rank-sum test and denoted as follows: ns, no significant; * $P < 0.05$; ** $P < 0.01$ (non-significant data in all comparisons are omitted). **b-c** Boxplot showing the concentrations of short-chain fatty acids (SCFAs) in the fecal (B) and serum (C) samples at baseline and post-treatment period. Boxes represent the interquartile range between the first and third quartiles and the median (internal line). Whiskers denote the lowest and highest values within 1.5 times the range of the first and third quartiles, respectively. Dots represent outlier samples beyond the whiskers. Wilcoxon rank-sum test. * $P < 0.05$; ** $P < 0.01$; *** $P < 0.001$. **d** The explained variations of fecal and serum SCFAs and clinical parameters by the gut microbiome at the post-treatment time point. Bar plots indicate the explained variation (effect size R^2) of each metabolite or parameter. **e** Network view of gut species, fecal and serum SCFAs, and clinical parameters. Circles represent the SCFAs or clinical parameters, while the surrounding connected triangles represent the gut species that had the highest contributions to the SCFAs or parameters and were used in the random forest models. Patients in the placebo, positive control, BL-99_low, and BL-99_high groups were administered with maltodextrin (2 g/day), rabeprazole (10 mg/day), low-dose BL-99 (1×10^{10} CFU/day), and high-dose BL-99 (5×10^{10} CFU/day), respectively.

Question 17: The stool microbiome shotgun studies and metabolomics however may not reflect

proximal actions of the probiotic, or changes at the mucosal associated microbiome (MAM) anywhere along the GI tract. The lack of upper GI biopsies with MAM results is a major limitation in interpreting this work.

Response: Thank you very much for your advice. MAM in the upper gastrointestinal tract may indeed be the pathophysiology of FD¹. However, this study focuses on the prolonged efficacy of BL-99 on symptom improvement in FD patients and its effects on fecal microbiota and metabolites. We found that after BL-99 treatment, the symptoms of FD patients improved while the fecal microbiota changed, and the SCFA-producing microbiota increased. Correspondingly, SCFA in stool and serum, and serum gastrin content increased. At the same time, SCFA infusion results showed that acetate and butyrate could stimulate serum gastrin level in SD rats (**Supplementary Fig. 5**). This directly proves that metabolites of gut microbiota can affect digestive function, providing solid evidence that the lower gastrointestinal (fecal) microbiota affects FD. Moreover, Wauters, L. et al. also demonstrated that *B coagulans* MY01 and *B subtilis* MY02 were efficacious in the treatment of FD, and this efficacy was associated with the abundance of *Faecalibacterium* in feces². It's concluded that BL-99 may improve FD symptoms by regulating the fecal microbiota.

The lack of upper GI biopsies with MAM results is a major limitation in interpreting this work. We have added related information in the Discussion section (Page 9, lines 249-253): 'Thirdly, considering the participants' wishes, we did not perform an endoscopy to collect gastroduodenal biopsies for mucosal-associated microbiome (MAM) detection. However, as probiotics mainly colonize the cecum and colon, we hypothesized that it exerts health effects primarily by regulating gut microbiota, which was also confirmed by our results.'

Supplementary Fig. 5. Effect of short-chain fatty acid infusion on serum gastrin. a acetate. b butyrate.

Significant differences among different groups were evaluated by one-way analysis of variance (ANOVA) with least significant difference (LSD) analysis, and the level of significance was set at * $P < 0.05$, ** $P < 0.01$, *** $P < 0.001$ vs. 2 $\mu\text{mol}/(\text{kg}\cdot\text{min})$ acetate or 0.1 $\mu\text{mol}/(\text{kg}\cdot\text{min})$ butyrate).

Reference

1. Ford, A. C. et al. Functional dyspepsia. *The Lancet*. **396**, 1689-1702 (2020).
2. Wauters, L. et al. Efficacy and safety of spore-forming probiotics in the treatment of functional dyspepsia: a pilot randomised, double-blind, placebo-controlled trial. *Lancet Gastroenterol. Hepatol.* **6**, 784-792 (2021).

We would again like to thank the reviewer for the careful reading and the helpful guidance about how to improve our study.

Reviewer #2 (Remarks to the Author):

In this manuscript (NCOMMS-23-04238), the authors reported prolonged efficacy of *B. lactis* in the treatment of functional dyspepsia (FD), and concluded that restoration of dysbiotic microbiota in the gut is an underlying mechanism of the alleviation of symptoms in FD patients treated with a probiotic strain.

Response: We thank the reviewer for a detailed assessment of our work and for the useful comments provided, which have helped us revising our work.

Comments:

Question 1: Although significant link of gut microbiota has been considered to be involved in the pathophysiology of functional GI disorders (FGID), almost of them have been about irritable bowel syndrome (IBS), not about FD (eg, Wauters et al. *Gut* 2020;69:591). With regard to the microbiota in the pathophysiology underlying FD, small intestinal bacterial overgrowth (SIBO) and gastroduodenal dysbiosis appear to attract much attention in the investigation of FD so far.

However, in the authors' study, gut microbiome and their bacterial metabolites were exclusively examined, but no analysis about microbiomes of stomach, duodenum or proximal small intestine was done; no consideration about them was found even in Discussion. Thus, the scope of the authors' study seems so deviated. In order to stress the importance of gut microbiota in the pathophysiology of FD, authors cited Reference No. 21 (*Gut* 2018;67:778) and mentioned in the text that "there was significant changes of gut microbiota composition in FD patients". But this referred article described about IBS patients but not mentioned about FD patients at all. Nevertheless, it would have been very interesting to know the authors' opinion how do a wrong change in the gut microbiota and their metabolites distort the function of stomach and duodenum in FD, which are anatomically away from distal small and large intestines in which a lot of gut microbiota colonize.

Response: Thanks for your valuable suggestions. The pathophysiology of functional dyspepsia (FD) is complex and heterogeneous, and its underlying mechanisms remain incompletely understood¹. As you have mentioned, small intestinal bacterial overgrowth (SIBO) and gastroduodenal dysbiosis have been proposed as potential pathophysiological mechanisms of FD². The purpose of this study is to explore the improvement effect of probiotic BL-99 on FD, rather than analyzing the pathogenesis of FD. The clinical experimental data obtained in this study clearly shows that probiotics can significantly improve FD by regulating the lower gastrointestinal (fecal) microbiota and metabolites. Wauters, L. *et al.* also demonstrated that the efficacy of probiotics against FD is related to the abundance of *Faecalibacterium* in feces³. Therefore, referring to previous studies, this study also explored the correlation between probiotics, fecal microbiota and metabolites, and FD.

In addition, thanks very much for pointing the misquotation of reference No. 21, and we have deleted it in the revised manuscript.

Reference

1. Ford, A. C. et al. Functional dyspepsia. *The Lancet*. **396**, 1689-1702 (2020).
2. Wauters, L. et al. Novel concepts in the pathophysiology and treatment of functional dyspepsia. *Gut*. **69**, 591-600 (2020).
3. Wauters, L. et al. Efficacy and safety of spore-forming probiotics in the treatment of functional dyspepsia: a pilot randomised, double-blind, placebo-controlled trial. *Lancet Gastroenterol. Hepatol*. **6**, 784-792 (2021).

Question 2: In this study, a significant beneficial clinical response was obtained in the probiotic-treated group when compared with the placebo-treated group. In addition, probiotic treatment in FD patients converted dysbiotic microbiome into those colonizing the gut of healthy subject. Although there is a coincidence of those events, it exists little evidence indicating causal relationship between them. While the authors indicated a significant correlation between SCFA/serum index and gut microbiome (Fig. 6), it will make little sense to demonstrate evidence that gut microbiota is really involved in probiotic-induced improvement FD symptoms in their study.

Response: Thank you very much for your advice. We have reorganized and mapped the metagenomic and metabolome sections, supplemented with additional trials to explore the role of probiotic in regulating fecal microbiota to improve FD. The results showed that, after BL-99 treatment, the symptoms of FD patients improved while the fecal microbiota changed, and the SCFA-producing microbiota of *Bifidobacterium animalis*, *Faecalibacterium prausnitzii*, and *Roseburia intestinalis* increased (**Fig. 2**). Correspondingly, SCFAs in stool and serum increased (**Fig. 3b**), and serum gastrin content increased (**Table 3**). Random forest models and contribution analysis of fecal microbiota and metabolites showed that the changes of FD clinical parameters were associated with the abundance of *Bifidobacterium animalis*, *Faecalibacterium prausnitzii*, and *Roseburia intestinalis* (**Fig. 3e**). To demonstrate the correlation between SCFAs and serum index, SCFAs were perfused into SD rats, and the results showed that acetate and butyrate could directly stimulate serum gastrin level (**Supplementary Fig. 5**). Moreover, SCFAs produced by gut microbiota have been confirmed to stimulate gastric G cells to secrete gastrin by activating parasympathetic nerves¹⁻³. In conclusion, BL-99 may improve FD symptoms by regulating fecal microbiota.

Fig. 2. Comparative analysis of the gut microbial composition in fecal samples from patients with functional dyspepsia (FD) treated with BL-99. **a** Principal coordinate analysis (PCoA) of microbiota communities in the fecal samples among four groups at baseline and post-treatment period. Samples are shown at the first and second principal coordinates (PCoA1 and PCoA2), and the ratio of variance contributed by these two PCoAs is shown. Ellipsoids represent a 95% confidence interval surrounding each group. The below and left boxplots show the sample scores in PCoA1 and PCoA2 (boxes show medians/quartiles; error bars extend to the most extreme values within 1.5 interquartile ranges). **b** Boxplot showing the relative abundances of *Bacteroidetes* and *Firmicutes* in samples at baseline and post-treatment period. Boxes represent the interquartile range between the first and third quartiles and the median (internal line). Whiskers denote the lowest and highest values within 1.5 times the range of the first and third quartiles, respectively. Dots represent outlier samples beyond the whiskers. Wilcoxon rank-sum test. * $P < 0.05$; ** $P < 0.01$; *** $P < 0.001$. **c** Changes in the abundance of species from the baseline to the post-treatment period. Heatmap shows the changes in the mean relative abundance of species from the baseline to the post-treatment period in samples within each group. For each species in each group, the significance levels of the comparisons between the changes in one group relative to the other three groups are calculated using the Wilcoxon rank-sum test and denoted as follows: ns, non-significant; * $P < 0.05$; ** $P < 0.01$ (non-significant data in all comparisons are omitted). **d** Changes in microbial functions from the baseline to the post-treatment period. **e** The relative abundance of *Bifidobacterium animalis* among the four groups at the follow-up period. Wilcoxon rank-sum test. * $P < 0.05$; ** $P < 0.01$; *** $P < 0.001$. Patients in the placebo, positive control, BL-

99_low, and BL-99_low groups were administered with maltodextrin (2 g/day); rabeprazole (10 mg/day); low-dose BL-99 (1×10^{10} CFU/day), and high-dose BL-99 (5×10^{10} CFU/day), respectively.

Fig. 3. Effects of BL-99 treatment on the fecal and serum metabolites in patients with functional dyspepsia. **a** Changes in fecal metabolites from the baseline to the post-treatment period. Heatmap shows the changes in the mean relative abundance of metabolites from the baseline to the post-treatment period in samples within each group. For each species in each group, the significance levels of the comparisons between the changes in one group relative to the other three groups are calculated using the Wilcoxon rank-sum test and denoted as follows: ns, no significant; * $P < 0.05$; ** $P < 0.01$ (non-significant data in all comparisons are omitted). **b-c** Boxplot showing the concentrations of short-chain fatty acids (SCFAs) in the fecal (B) and serum (C) samples at baseline and post-treatment period. Boxes represent the interquartile range between the first and third quartiles and the median (internal line). Whiskers denote the lowest and highest values within 1.5 times the range of the first and third quartiles, respectively. Dots represent outlier samples beyond the whiskers. Wilcoxon rank-sum test. * $P < 0.05$; ** $P < 0.01$; *** $P < 0.001$. **d** The explained variations of fecal and serum SCFAs and clinical parameters by the gut microbiome at the post-treatment time point. Bar plots indicate the explained variation (effect size R^2) of each metabolite or parameter. **e** Network view of gut species, fecal and serum SCFAs, and clinical parameters. Circles represent the SCFAs or clinical parameters, while the surrounding connected triangles represent the gut species that had the highest contributions to the SCFAs or parameters and were used in the random forest models. Patients in the placebo, positive control, BL-99_low, and BL-99_high groups were administered with maltodextrin (2 g/day), rabeprazole (10 mg/day), low-dose BL-99 (1×10^{10} CFU/day), and high-dose BL-99 (5×10^{10} CFU/day), respectively.

Supplementary Fig. 5. Effect of short-chain fatty acid infusion on plasma gastrin. a acetate. b butyrate.

Significant differences among different groups were evaluated by one-way analysis of variance (ANOVA) with least significant difference (LSD) analysis, and the level of significance was set at * $P < 0.05$, ** $P < 0.01$, *** $P < 0.001$ vs. 2 $\mu\text{mol}/(\text{kg}\cdot\text{min})$ acetate or 0.1 $\mu\text{mol}/(\text{kg}\cdot\text{min})$ butyrate).

Table 3. Within-group changes in functional dyspepsia - relative serum indexes

Change in serum indexes ^{a/} (ng/mL)		Placebo (n = 45)	Positive_control (n = 48)	BL-99_low (n = 47)	BL-99_high (n = 45)
From the baseline to the post-treatment period	PGI ^{b)}	3.39 ^{ab} (-3.16 to 9.94)	14.00 ^a (6.25 to 21.74)	2.79 ^b (-7.15 to 12.73)	10.58 ^{ab} (3.88 to 17.29)
	PGII ^{b)}	-0.45 ^a (-2.25 to 1.64)	-2.38 ^a (-5.59 to 0.82)	-0.61 ^a (-1.80 to 0.59)	-2.24 ^a (-4.09 to -0.39)
	PGR ^{c)}	0.85 ^b (-0.58 to 2.27)	4.59 ^a (2.77 to 6.40)	0.72 ^b (-0.21 to 1.65)	2.23 ^b (-0.02 to 4.48)
	G17 ^{d)}	0.14 ^b (-0.39 to 0.68)	0.78 ^b (-0.10 to 1.67)	1.87 ^a (0.91 to 2.83)	4.11 ^a (2.62 to 5.61)
From baseline to the 2-week follow-up period	PGI	-2.82 ^a (-7.65 to 2.01)	-3.02 ^a (-8.84 to 2.79)	-1.31 ^a (-6.24 to 3.62)	-2.15 ^a (-6.25 to 1.96)
	PGII	-1.26 ^{ab} (-3.21 to 0.68)	-3.44 ^a (-4.49 to -2.40)	-1.18 ^b (-3.14 to 0.79)	-0.64 ^b (-2.39 to -1.12)
	PGR	-0.18 ^b (-1.30 to 1.67)	2.68 ^a (1.41 to 3.95)	0.77 ^b (-0.43 to 1.96)	-0.54 ^b (-1.85 to 0.77)
	G17	-0.47 ^a (-3.06 to 2.11)	0.11 ^a (-0.96 to 1.17)	0.35 ^a (-1.04 to 1.74)	0.33 ^a (-1.59 to 2.25)

Data are estimates (95% confidence interval) in the per-protocol (PP) set.

BL-99, *Bifidobacterium animalis* subsp. *Lactis* BL-99

Patients in the placebo, positive control, BL-99_low, and BL-99_high groups were administered with maltodextrin (2 g/day), rabeprazole (10 mg/ day), low-dose BL-99 (1×10^{10} CFU/day), and high-dose BL-99 (5×10^{10} CFU/day), respectively.

^{a)}PGI: pepsinogen I; ^{b)}PGII, pepsinogen II. ^{c)}PGR: pepsinogen ratio = PGI/PGII. ^{d)} G17: Gastrin 17.

^{*)} One-way analysis of variance with least significant difference method was used to analyze the differences among the four groups. Differences were considered significant at $P < 0.05$ and are presented with different characters.

Reference

1. Engevik, A. C. et al. The physiology of the gastric parietal cell. *Physiol. Rev.* **100**, 573-602 (2020).
2. Margolis, K. G. et al. The microbiota-gut-brain axis: From motility to mood. *Gastroenterology (New York, N.Y. 1943)*. **160**, 1486-1501 (2021).
3. Perry, R. J. et al. Acetate mediates a microbiome-brain-beta-cell axis to promote metabolic syndrome. *Nature*. **534**, 213-217 (2016).

Question 3: Although the authors measured serum PGI, PGII and G17 before and after probiotic treatment and presented the within-group change in FD patients (Table 2). However, there was no data indicating some significant difference in the values of those parameters between FD patients and healthy control subjects without probiotic treatment in their study. Without such fundamental evidence or some supporting citations, why were the authors able to imply involvement of those serum activities in the alleviation of FD symptoms.

A change in serum PGI and PGII by administration of probiotics are already reported (Aliment Pharmacol Ther 2006;23:1077).

Response: Thank you for your question. Studies have confirmed that serum pepsinogen and gastrin levels in FD patients are different from those in healthy individuals and are associated with various symptoms of FD^{1,2}. The Maastricht V/Florence Consensus Report states that pepsinogen and gastrin are important parameters for managing FD³. Although we did not test the level of serum pepsinogen and gastrin in healthy control subjects in our study, the value of these parameters in FD patients before probiotics treatment was comparable to values reported in previous studies involving FD patients. Moreover, Igarashi et al. found that *Lactobacillus gasseri* OLL2716 (LG21) increased serum PGI levels while alleviating FD symptoms⁴. Kwon et al. reported that different treatment methods increased serum G17 levels in FD patients⁵. Overall, pepsinogen and gastrin can be used as objective indicators to characterize FD symptoms. Therefore, we focused on these two indicators. We also made relevant additions in the Discussion section (Page 7, lines 204-209): 'Studies have confirmed that serum pepsinogen and gastrin levels in FD patients are different from those in healthy persons and are associated with various symptoms of FD. The Maastricht V/Florence Consensus Report states that pepsinogen and gastrin are important parameters for managing FD. Furthermore, Igarashi et al. found that *Lactobacillus gasseri* OLL2716 improved FD symptoms and altered serum PGI levels. Therefore, PGI, PGII, and G17 were determined in this study.'

Reference

1. Crafa, P. et al. Functional dyspepsia. *Acta Biomed.* **91**, e2020069 (2020).
2. Tahara, T. et al. Examination of serum pepsinogen in functional dyspepsia. *Hepatogastroenterology.* **59**, 2516-2522 (2012).
3. Malfertheiner, P. et al. Management of *Helicobacter pylori* infection—the Maastricht V/Florence Consensus Report. *Gut.* **66**, 6-30 (2016).
4. Igarashi, M. et al. Correlation between the serum pepsinogen I level and the symptom degree in proton Pump inhibitor-users administered with a probiotic. *Pharmaceuticals.* **7**, 754-764 (2014).
5. Kwon, C. et al. Acupuncture as an add-on treatment for functional dyspepsia: A systematic review and meta-analysis. *Front. Med.* **8**, 682783 (2021).

Question 4: There are some wrong citations. For example, the authors mentioned in detail (Lines 214-218) that “Some scholars compared,,24 FD patients and 21 age-matched,,in the control group.” Although that citation appears very important to support the authors’ opinion, no article was referred for that.

The reason why Reference No. 21 is inappropriate is already shown in Comment 1.

Response: We thank the reviewer for highlighting this issue. We have deleted reference No. 21 and revised the description in the Discussion section (Page 8, lines 216-226): 'Although the pathogenesis of FD has not been elucidated, Wauters L *et al.* demonstrated that *B. coagulans* MY01 and *B. subtilis* MY02 exerted potent therapeutic effects on FD by modulating the abundance of fecal *Faecalibacterium*¹³. Based on previous studies, this study examined the correlation between probiotics, fecal microbiota and metabolites, and FD. The results of this study suggested that BL-99 intervention alleviated the symptoms of FD, altered the fecal microbiota composition, and upregulated the abundance of SCFA-producing microbiota. Random forest models and contribution analysis of fecal microbiota and metabolites revealed that the alleviation of FD symptoms was dependent on the abundance of *Bifidobacterium animalis*, *Faecalibacterium prausnitzii*, and

Roseburia intestinalis. In summary, BL-99 alleviated the symptoms of FD, altered the composition of fecal microbiota, and upregulated the abundance of SCFA-producing microbiota.'

We would like to again thank the reviewer for the helpful guidance about how to improve our study.

Reviewer #3 (Remarks to the Author):

There is some useful information about the impact of probiotics for this condition in this population.

However, there is a lack of clarity around some of the methods and data analysis, especially with regards the primary outcome of the clinical trial which means I would not recommend publication. There is no acknowledgement of the disadvantages of a per-protocol analysis, nor the subjectivity of the outcome given no blinding in the trial.

The methods section needs much more detail as outlined below, as there is currently not enough detail to replicate the methods or to aid understanding of the impact of the intervention on clinical symptoms.

Response: Thanks for your insightful and constructive suggestions, which have been invaluable for improving our article. We cannot agree more that it is crucial to describe sufficient detail of the study design, implementation process, and data analysis to enable the reader to repeat the study process and to aid the understanding of the results. We acknowledge that our previous manuscript fell short of your expectations, and we apologize for any disappointment caused. We have thoroughly revised and polished the article based on your valuable suggestions. Specific modifications and replies are as follows.

Question 1: Abstract: The primary outcome should be identified in the abstract in the methods section. Also it should be outlined how the clinical response rates were measured, so the results section needs to be reduced to ensure this information can be included. In the results section I would also suggest indicating that there was no evidence for a difference between the BL-99 high group and BL-99 low group after the 8 week follow up period.

Response: Thank you for your suggestions. We have indicated that the primary outcome of this study was clinical response rate (CRR) of FD score after 8-week treatment and outlined the calculation method of CRR in the Abstract section (Page 1, lines 9-11): 'The primary outcome was the clinical response rate (CRR) of FD score after 8-week treatment, defined as the proportion of participants with a decrease >0.5 .'

Furthermore, symptoms result at the 2-week follow-up and 8-week questionnaire survey were summarized in the Abstract section (Page 1, lines 13-14): 'This effect was sustained until 2-week after treatment but disappeared 8-week after treatment.'

Question 2: In general, for transparency - it should be written somewhere in the manuscript who sponsored and funded this clinical trial. I would also suggest the clinical trials registration information goes in the abstract but that may not be common in this journal. As a minor point - there were also a couple of typos and one unfinished sentence.

Response: Thank you for your reminding. The information on the sponsorship and funding for this study was provided in the Acknowledgements section of the revised manuscript (Page 22, lines 599-603): 'This work was financially supported by the National Key R&D Program of China (grant number 2021YFD1600204), The 111 project from the Education Ministry of China (No. B18053), and the Key project of multidisciplinary Clinical Research Innovation Team of Beijing Chaoyang Hospital Affiliated to Capital Medical University (CYDXK202207).'

In accordance with your suggestion, we have supplemented the clinical trial registration

information in the Abstract section, as a custom in this journal (Page 1, lines 6-9): 'This randomized controlled clinical trial (Chinese Clinical Trial Registry, ChiCTR2000041430) assigned 200 FD patients to receive placebo, positive-drug(rabeprazole), or *Bifidobacterium animalis* subsp. *Lactis* BL-99(BL-99; low, high doses) for 8-week.'

Furthermore, we have had the entire manuscript carefully reviewed and refined by a professional English language expert. (Proofreading Certificate No.: ASLESTD0202937 and Editorial Certificate Code: 5e1fa00e8d0abd056d261c973fe81451).

Question 3: Introduction: There is an incorrect citation. 'Global prevalence estimate' comes from the following paper (AC Ford, A Marwaha, R Sood, P Moayyedi. Global prevalence of, and risk factors for, uninvestigated dyspepsia: a meta-analysis. *Gut*, 64 (2015), pp. 1049-1057) not reference 2 which simply refers to the paper mentioned above.

Response: Thank you for bringing this to our attention. We carefully reviewed the reference in the Introduction section. Indeed, it was an incorrect citation. This mistake has been corrected in the revised manuscript (Page 2, lines 20-25): 'The global prevalence of uninvestigated dyspepsia is 21%. Epidemiological investigations have revealed that the symptoms vary in approximately two-thirds of patients with FD irrespective of postprandial distress syndrome (PDS; with postprandial fullness and early satiety symptoms) or epigastric pain syndrome (EPS; with epigastric pain and epigastric burning symptoms) subtypes. In patients with FD, these symptoms are persistent for at least one to three days per week and last for more than 3 months.'

Additionally, we have conducted a thorough review and refinement of all references throughout the manuscript.

Question 4: Results: It is important to list the 'other' reasons for exclusion, in particular for those that were enrolled but excluded prior to randomisation. More information should be given about what was done at the beginning of the 2-week run in period and what was done at the start of treatment. Were the same questionnaires used at every visit? This is not clear.

Response: Thanks for your suggestion. We have listed 'other reasons' in Figure 1 as a footnote (Page 16, lines 538-539): '**Fig. 1.** Trial profile. Other reasons include participant relocation, unable to be contacted, and time conflicts.'

In addition, we updated the implementation details of the entire study process, including the run-in, baseline, visit, and questionnaire survey periods, as well as the questionnaire used in each visit or survey (Pages 10-11, lines 313-325): 'All included participants first underwent a 2-week run-in period. During the run-in period, participants were not allowed to take foods containing probiotics (such as probiotic powder, probiotic yogurt, etc.). Then participants were treated with PPI, BL-99, or placebo for 8 weeks, followed by an 8-week post-treatment follow-up. During the treatment, participants were instructed to maintain their habitual lifestyle habits such as diet and physical activity and were not allowed to take antibiotics. A total of 4 visits [at baseline (V1), 4-week treatment (V2), 8-week treatment (V3), and 2-week follow-up (V4)] and 1 survey [questionnaire surveys 8 weeks after the treatment (V5)] were conducted throughout the study period. At each visit and the final survey, participants were surveyed using a uniform FD symptom assessment questionnaire (see the 'Symptom assessment' section for details). Blood and fecal samples were collected at V1, V3, and V4. Even 8 weeks post the treatment, we were fortunate to receive the questionnaire responses from all participants who completed the treatment.'

Question 5: It is difficult to understand the 'average decrease' value as a clinical response rate. The average decrease should be an absolute number per participant and so doesn't link well into the clinical response rate which is a percentage of the group having a decrease above a certain level. I am not sure but I think if you are saying what proportion in each group had a decrease in their EPS+PDS score of >0.5 then that is not the 'average'. Elsewhere in the publication the word 'average' is not used which I think is better.

Response: Thanks for your comments. In the original manuscript, the connection between 'average decrease' and 'clinical response rate' was achieved through the following process. Firstly, we averaged the scores of four FD symptoms as a composite FD score for each participant (namely, the 'total score' in **Table 1** of the original manuscript). Then, a decrease > 0.5 in this average score was defined as clinical response. Finally, clinical response rate was calculated as the proportion of clinical responders. We apologize for the confusion caused by our insufficient explanation in the original manuscript. We revised the 'total score' as 'FD score' in the revision, along with further improvements in definitions and descriptions (Page 11, lines 338-342): 'Specifically, FD symptoms included postprandial fullness, early satiety, epigastric pain, and epigastric burning, with a score range of 0-3 for each symptom (0 = none, 1 = mild, 2 = moderate, and 3 = severe). FD score was calculated as the average of the four symptoms. PDS score was the average score of postprandial fullness and early satiety, and the EPS score was the average score of the remaining two symptoms.'

Question 6: I think it would also be important to present the mean (sd) of the scores individually (i.e PDS and EPS) and combined (PDS+EPS) at 8 weeks. Also the combination of PDS+EPS scores should be presented in the baseline table (Table 1).

Response: Thanks for your suggestion. We have supplemented the mean (sd) of PDS, EPS, and FD scores of the 4 treatment groups for all periods (including baseline, 4-week treatment, 8-week treatment, 2-week follow-up, and 8-week questionnaire survey) (**Supplementary Table 6 and Table 7**). And the results were shown in the revised manuscript (Page 4, lines 100-103): 'In addition, The means of FD, PDS, and EPS scores were also compared among the four groups (**Supplementary Table 6, Table 7**). The results showed that there were no significant differences in FD, EPS, or PDS scores between the 4 treatment groups at any visit or survey in either the ITT or PP analyses (all P_{overall} values are ≥ 0.05).'

The 'combination of PDS+EPS scores' corresponds to 'total score' in **Table 1** of the original manuscript, which we have renamed as the 'FD score' in the revised manuscript for better clarity (Page 11, lines 338-342): 'Specifically, FD symptoms included postprandial fullness, early satiety, epigastric pain, and epigastric burning, with a score range of 0-3 for each symptom (0 = none, 1 = mild, 2 = moderate, and 3 = severe). FD score was calculated as the average of the four symptoms. PDS score was the average score of postprandial fullness and early satiety, and the EPS score was the average score of the remaining two symptoms.'

Question 7: Also in Table 1 it would be good to understand how the Total score was calculated. It does not appear to be the addition of clinical symptoms for all the symptoms.

Response: Thanks for your reminding. We apologize for any confusion caused by the previous description. We have renamed the 'total score' as the 'FD score' for better clarity, and have provided a detailed definition and calculation method in the revised manuscript (Page 11, lines 338-342):

'Specifically, FD symptoms included postprandial fullness, early satiety, epigastric pain, and epigastric burning, with a score range of 0-3 for each symptom (0 = none, 1 = mild, 2 = moderate, and 3 = severe). FD score was calculated as the average of the four symptoms. PDS score was the average score of postprandial fullness and early satiety, and the EPS score was the average score of the remaining two symptoms.'

Question 8: Baseline characteristics across groups should not be compared as the randomisation is designed to give similar distributions so the footnote regarding this should be removed from Table 1.

Response: Thanks for your suggestion. We have revised **Table 1** by removing the between-group comparison results and their corresponding footnotes.

Question 9: In the legend to the table S1 and S2 and Figure 2 it should be outlined what significant level each letter corresponds without needing to refer back to the main text. I also think that the values for Figure 2 should be reported in a table, not just the different values for male and females as is currently in the appendix.

Response: Thank you for your suggestion. We have updated original Tables S1 and S2 by adding corresponding significance levels, which are renumbered as **Supplementary Table 2 and Table 3** in the revised version of materials. Additionally, the values and corresponding significance levels for the original Figure 2 have been reported in a newly created **Table 2**, with the original Figure 2 deleted.

Question 10: Importantly it is not clear why the numbers in each group in Table S1 for the primary outcome do not match those included in the analysis in Figure 1.

Response: Thanks for your reminding. It should be noted that **Fig. 1** displays the total number of participants in each group, including both males and females. Tables S1 and S2 (which have been renumbered as **Supplementary Table 2 and Table 3** in the revised version of materials), on the other hand, provide the number of male and female participants in each group, respectively. We have thoroughly cross-checked the participant numbers in **Fig. 1, Supplementary Table 2, and Table 3**, and have confirmed their accuracy.

Question 11: There is mention of a per-protocol analysis but no definition in the text of how per-protocol was defined. It is not sufficient to have this just in the SAP and it should be outlined in the methods section.

Response: Thanks for your reminding. We have added a definition of the per-protocol (PP) set analysis in the method (Page 13, lines 381-386): 'The main data set for efficacy analysis in this study was PP set, which referred to participants that had completed the planned treatment and visits according to the protocol and had no obvious effect on the therapeutic effect. Violations that significantly affect efficacy included (but were not limited to) the following: a. Received interference therapy after inclusion; b. Poor compliance (e.g., with follow-up visits less than 80% of the required number of visits); c. Follow-up beyond the window period.'

Discussion:

Question 12: There needs to be acknowledgement of the subjectivity of patient reported clinical symptoms and using symptom scores. This was not a blinded trial so all participants had knowledge of their allocation, or if it was blinded then there are not details of the blinding in this manuscript. Thus, assuming no blinding, those getting the strongest intervention potentially knew they were getting it and may report lower clinical symptom scores.

Response: Thanks for your reminding. Sorry for the ambiguity about blinding in the previous manuscript. We have added the details of blinding in the Methods section as follows (Page 10, lines 304-310): 'Due to the difficulty of making probiotic formulations identical to PPI drugs, blinding was not possible in all four treatment groups. The positive-drug group was treated with PPI pills, and the other three groups received solid beverage powder with identical appearance, taste, and smell between groups. Therefore, the positive-drug group was open-label. For the other three groups, researchers and participants were blinded to treatment assignments until the study was completed.'

We fully agree with you that participants' subjectivity is an important factor affecting the accuracy of symptom scoring. Therefore, we have supplemented a discussion of the influence of our blind design on the results in the limitation section. (Page 9, lines 253-260): 'Fourthly, in this study, the positive-drug group was not blinded, so the participants' subjectivity may affect the accuracy of symptom reporting, which may bias the results. However, since the double-blind method was successfully implemented in the BL-99_high, BL-99_low, and placebo groups, the effect of BL-99 relative to placebo should be credible, and the results of these 3 groups also confirmed this. In addition, even if the positive-drug group was not blinded, the clinical response rate of the BL-99_high group was still higher than that of the drug group after the 8-week treatment, which further supports the conclusions of this study.'

Question 13: There also needs to be discussion of the limitation of per protocol analyses and the fact that this introduces bias that randomisation is trying to remove.

Response: Thanks for your suggestion. We fully agree with you that the per-protocol analyses have some limitations. In the revised manuscript, we additionally analyzed symptom scores according to the intention-to-treat (ITT) principle (**Supplementary Table 1**). For the primary outcome, the ITT analysis yielded similar results to those of PP, which also supported our main conclusion (Pages 3-4, lines 80-86): 'Similar results were observed in the intention-to-treat (ITT) set, which are presented in **Supplementary Table 1**. It also showed that the high dose of BL-99 group [45 (90.0%)] had significantly higher 8-week-treatment CRR of FD score compared to placebo [29 (58.0%), $P = 0.001$], BL-99_low [37 (74.0%), $P = 0.044$] and positive control group [35 (70.0%), $P = 0.017$]. The results for post-treatment 2-week and post-treatment 8-week CRR of FD score in the ITT analysis were also similar to that of the PP analysis.'

Question 14: There is no justification of the choice of endpoint and why scores from PDS and EPS were combined and thus that needs to be added to the discussion section.

Response: Thank you for your reminding. The primary outcome of our study is clinical response rate, defined as the proportion of participants in the group with a decrease in FD score of greater than 0.5 after 8-week treatment. We chose this primary outcome based on a study conducted at University Hospitals Leuven that also evaluated the efficacy of probiotics in improving FD^{1,2}. Regarding the combination of PDS and EPS scores as FD score, we referred to studies in which the FD score represented a comprehensive evaluation of FD symptoms³. We have

included these details in the Discussion section (Page 7, lines 183-189): 'The CRR of FD score after 8-week treatment was chosen as the primary outcome in our study, which was based on a study conducted in University Hospitals Leuven that also evaluated the efficacy of probiotics in improving FD. At the same time, a previous FD questionnaire validation study confirmed that a change of 0.5 was the threshold for the minimal clinically significant difference. In addition, referring to similar studies, we combined PDS and EPS score as FD score, which represented a comprehensive evaluation of FD symptoms.'

Reference

1. Carbone, F. et al. Validation of the Leuven Postprandial Distress Scale, a questionnaire for symptom assessment in the functional dyspepsia/postprandial distress syndrome. *Aliment. Pharmacol. Ther.* **44**, 989-1001 (2016).
2. Wauters, L. et al. Efficacy and safety of spore-forming probiotics in the treatment of functional dyspepsia: a pilot randomised, double-blind, placebo-controlled trial. *Lancet Gastroenterol. Hepatol.* **6**, 784-792 (2021).
3. Kamiya, T. et al. A multicenter randomized trial comparing rabeprazole and itopride in patients with functional dyspepsia in Japan: the NAGOYA study. *J. Clin. Biochem. Nutr.* **60**, 130-135 (2017).

Methods:

Question 15: It should be made clear it was run at two hospital sites and not one (as the first sentences infer) and potentially give recruitment numbers per group per site in Table 1.

Response: Thanks for your reminding. We have revised the statement to clarify that the study was conducted in two hospitals (Page 10, lines 286-288): 'Outpatients (18-60 years) with FD symptoms were recruited and screened between 26 December 2020 and 10 February 2021 at Beijing Chaoyang Hospital, Capital Medical University (CCMU), and Chinese PLA General Hospital (CPLAGH).'

Additionally, we have added the number of participants recruited from each hospital in **Table 1**.

Question 16: As a minor point the word 'human' is not necessary before the word trial. That is made clear from the beginning.

Response: Thanks for your advice. We have removed the word 'human' in the revised manuscript.

Question 17: As mentioned in the results section the definition of clinical response rate needs to be linked to the fact that it is reported as a percentage. So it needs to have a denominator and numerator so if it is the number that have the decrease > 0.5 then it should be written as such.

Response: Thanks for your carefulness. We apologize for any confusion caused by our previous manuscript. We have clarified the definition and calculation method of clinical response rate in the revised manuscript (Page 12, lines 359-361): 'The primary outcome was the clinical response rate (CRR) of the FD score at week 8 of treatment. Clinical response was defined as a score (i.e., FD score, PDS score, and EPS score) decrease > 0.5. CRR was then calculated as the proportion of clinical responders.'

Question 18: Under the sample size section it should be made more explicit what information was used from reference 16 to input into the sample size calculation. It should be clear if the comparison was made on the percentages across the arms and if so what percentages were the assumptions based

on (was the difference hypothesised to be the same for each group compared to placebo?). Also, there is no consideration of multiple testing and that three arms are being tested against the placebo.

Response: Thanks for your reminding. We apologize for the lack of clarity in the sample size calculation process in the original manuscript. We have described the calculation process in detail in the revised manuscript. As for multiple testing, we didn't take this into account in the calculation. The details are as follows.

(Page 12, lines 372-380): 'In a study of probiotics improving FD, the CRRs after 8 weeks of treatment of probiotic (5×10^9 CFU/day) and placebo were 48% and 20%, respectively, which were thus assumed for the low-dose probiotic group (1×10^{10} CFU/day, 48%) and the placebo group (20%) in our sample size calculation. In addition, we assumed a CRR of 50% for the positive drug group based on a study, and an intermediate value of 49% for the high-dose probiotic group, which was between the low-dose probiotic group and the positive drug group. Based on these assumptions, a sample size of 42 would be required per group (power of 80% and two-sided $\alpha = 0.05$). Considering a 20% dropout rate, 50 subjects would be needed to be included in each group (200 for 4 groups).'

Question 19: It needs to be made clear that there was not a visit at the end of 8 weeks after treatment finished and it was only a survey. Were the completeness rates different and if so this should be made clear.

Response: Thanks for your reminding. As you pointed out, we conducted only a symptom survey 8 weeks post-treatment, which was not a 'visit'. We have revised the word in the manuscript to 'survey' (Page 11, lines 318-321): 'A total of 4 visits [at baseline (V1), 4-week treatment (V2), 8-week treatment (V3), and 2-week follow-up (V4)] and 1 survey [questionnaire surveys 8 weeks after the treatment (V5)] were conducted throughout the study period.'

Regarding the completion rate of this survey, we were fortunate to receive responses from all participants who completed the treatment, resulting in a completion rate of 100%. This was addressed in the revised manuscript as well (Page 11, lines 323-325): 'Even 8 weeks post the treatment, we were fortunate to receive the questionnaire responses from all participants who completed the treatment.'

Question 20: There should be a statement about missing data in the main text of the methods section. There is a line referenced in the STORMS/STROBE checklist, it did not seem to be about missing data particularly for primary and secondary outcomes.

Response: Thank you for your reminding. We have included a statement on how missing data were handled in the revised manuscript (Page 13, lines 386-389): 'We also analyzed symptom scores based on the ITT set including all participants who were randomized. In the ITT set, missing values for symptom scores were imputed based on the last observation carried forward method.'

We would again like to express our gratitude to the reviewer for the excellent guidance about how to improve our study and manuscript. Many thanks!

REVIEWER COMMENTS

Reviewer #1 (Remarks to the Author):

The manuscript has been extensively revised and is considerably improved.

I remain concerned a per protocol analysis is presented as the primary outcome when an ITT is always the preferred (and conservative) analysis approach in clinical trials. I recommend the PP analysis go into the Supplementary and the ITT be in the main results section. The ITT results are less impressive but are more likely to reflect the reality of treatment efficacy. What was pre-specified in the protocol? This change will alter the results, and discussion somewhat.

I suggest adding in a summary of the data (as a post hoc analysis) on those who became symptom free on each treatment (and p values) to the results in the revised paper (as provided to the reviewers).

I'm still not convinced the gastrin data fit with colonic microbiome alterations as this study implies because there is no clear cut mechanism. I note the new preliminary animal work but the conclusions this provides "proof" needs toning down. Please add in more details of these experiments: details of the rats (SD means?), how many etc.

The Maastricht consensus refers to *H. pylori* positive patients in general. As this was a Hp negative cohort the relevance of referring to this consensus in supporting testing gastrin and pepsinogen testing seems irrelevant.

Reviewer #3 (Remarks to the Author):

The authors have taken care to address all comments made by reviewers, and this has led to a much improved manuscript.

Reviewer #4 (Remarks to the Author):

Summary: The current treatments for functional dyspepsia (FD) are not consistently effective. This study was conducted where 200 FD patients were given either a placebo, a known drug (rabeprazole), or different doses of a probiotic called *Bifidobacterium animalis* subsp. *Lactis* BL-99 for 8 weeks. The main measure was the clinical response rate (CRR) of FD, specifically if there was a decrease in FD score by more than 0.5. Results showed that the high dose of BL-99 had a 95.6% CRR, which was notably better than the placebo, low dose BL-99, and the known drug. This positive effect from BL-99 lasted for 2 weeks post-treatment but vanished by the 8th week after treatment. Further analysis indicated that BL-99 boosts the presence of SCFA-producing bacteria and raises SCFA levels in both stool and blood, which in turn affects the release of a hormone called gastrin. This suggests that BL-99 has potential as a sustainable treatment for FD relief.

The study is of interest but the results are not particularly novel and shows modest effects. Moreover, the study provides no clear convincing arguments for the mechanisms proposed to mediate the beneficial effect. Both the appearance of SCFA and the modulation of gastrin level had been reported before.

In detail:

The confounding effects of increased body weights was not assessed. Hence the population studied is not really representative, same applies to gender distribution.

No objective endpoint was applied to measure the effects observed.

The effect would have to be corroborated by using a different bacterial strain in order to prove that the effect is strain specific.

The differential expression of SCFA during probiotic treatment is rather expected and likely not strain specific.

The influence of SCFA on gastrin levels in humans and rodents is no proof of efficacy of this probiotic treatment.

There is no data on the evaluation of food frequency questionnaires in association with the changes in microbiota.

Were the methodologies applied all from the very same fecal isolate or were different portions of the stool samples used for analysis. If the entire metabolites are not taken and worked up for the entity of DAN, metabolites etc, an enormous bias can be expected.

Point-by-point response to the reviewers' comments

Reviewer #1 (Remarks to the Author):

The manuscript has been extensively revised and is considerably improved.

Question 1: I remain concerned a per protocol analysis is presented as the primary outcome when an ITT is always the preferred (and conservative) analysis approach in clinical trials. I recommend the PP analysis go into the Supplementary and the ITT be in the main results section. The ITT results are less impressive but are more likely to reflect the reality of treatment efficacy. What was pre-specified in the protocol? This change will alter the results, and discussion somewhat.

Response: Thanks for your suggestion. Following your suggestion, we have put the results of the ITT analysis in the main results section (Pages 3-4, lines 73-97) and the PP analysis in Supplementary Table 1. The details are as follows.

(Pages 3-4, lines 73-97): 'The results of the effect on clinical response rate (CRR) based on the intention-to-treat (ITT) set are presented in **Table 2**. The primary outcome, 8-week CRR of FD score was significantly higher for BL-99_high [45 (90.0%)] than placebo [29 (58.0%), $P = 0.001$], BL-99_low [37 (74.0%), $P = 0.044$] and positive_control group [35 (70.0%), $P = 0.017$]. At the 2-week follow-up after the treatment, the CRR of FD score in the BL-99_high group [42 (84.0%)] was still significantly higher than placebo [31 (62.0%), $P=0.016$] and positive_control group [33 (66.0%), $P=0.041$], but there was no significant difference between the BL-99_high and BL-99_low groups. Post-treatment follow-up at 8 weeks no longer showed significant differences in CRR between the 4 groups. Similar results were observed in the per-protocol (PP) set, which are presented in **Supplementary Table 1**. It also showed that the high dose of BL-99 group [43 (95.6%)] had a significantly higher 8-week-treatment CRR of FD score compared to placebo [28 (62.2%), $P = 0.001$], BL-99_low [36 (76.6%), $P = 0.019$] and positive_control group [34 (70.8%), $P = 0.006$]. The results for post-treatment 2-week and post-treatment 8-week CRR of FD score in the PP analysis were also similar to that of the ITT analysis.

Regarding the EPS score, 8-week CRR in the BL-99_high group [37 (74.0%)] was significantly higher than placebo [24 (48.0%), $P = 0.009$] and positive drug group [27 (54.0%), $P = 0.039$]. Even 8 weeks after the treatment, the BL-99_high group still had significantly higher CRR than the placebo and BL-99_low group. The results of the PP analysis were consistent with this.

As for the PDS score, the 4-week CRR in the BL-99_high group was consistently higher than the placebo and positive drug group in both the ITT and PP analyses. This difference between groups persisted at the 2-week follow-up after the treatment only in the PP analysis, but not in the ITT analysis. No significant differences between the four groups were observed at other study points (visit or survey).'

Question 2: I suggest adding in a summary of the data (as a post hoc analysis) on those who became symptom free on each treatment (and p values) to the results in the revised paper (as provided to the reviewers).

Response: Thank you very much for your advice. As suggested, we now provide a summary of the data (as a post hoc analysis) on those who became symptom-free on each treatment (and p values), which was shown in **Supplementary Table 6**. And we have added related results in the revised manuscript (Page 4, lines 103-109):

‘As a post hoc analysis, data on patients who became symptom-free after treatment were also presented based on the ITT set (**Supplementary Table 6**). It shows that the proportion of people with no FD symptoms was significantly higher in the BL-99_high group (39 [78.0%]) than that in the placebo (18 [36.0%], $P < 0.001$) and the positive_control (21 [42.0%], $P < 0.001$) groups at the 2-week follow-up after the treatment, but not at 8-week treatment. Similar results were observed for the proportion of those with no PDS symptom both at 8-week treatment and 2 weeks after the treatment.’

Supplementary Table 6 No symptoms after treatment for all (men and women) participants based on intention-to-treat (ITT) set

No symptoms		Placebo (n=50)	Positive_control (n=50)	BL-99_low (n=50)	BL-99_high (n=50)	P overall	P					
							Positive_control vs Placebo	BL-99_low vs Placebo	BL-99_high vs Placebo	BL-99_low vs Positive_control	BL-99_high vs Positive_control	BL-99_high vs BL-99_low
4-week treatment	FD score ^{a)}	18 (36.0)	13 (26.0)	16 (32.0)	21 (42.0)	0.431	-	-	-	-	-	-
	PDS score ^{b)}	21 (42.0)	21 (42.0)	22 (44.0)	26 (52.0)	0.280	-	-	-	-	-	-
	EPS score ^{c)}	34 (68.0)	25 (50.0)	27 (54.0)	30 (60.0)	0.106	-	-	-	-	-	-
8-week treatment	FD score	21 (42.0)	18 (36.0)	29 (58.0)	28 (56.0)	0.554	-	-	-	-	-	-
	PDS score	24 (48.0)	25 (50.0)	30 (60.0)	34 (68.0)	0.680	0.841	0.230	0.044	0.029	0.046	0.840
	EPS score	36 (72.0)	30 (60.0)	38 (76.0)	34 (68.0)	0.471	-	-	-	-	-	-
2-week follow-up	FD score	18 (36.0)	21 (42.0)	36 (72.0)	39 (78.0)	0.004	0.539	<0.001	<0.001	0.03	<0.001	0.489
	PDS score	21 (42.0)	22 (44.0)	40 (80.0)	39 (78.0)	<0.001	0.840	<0.001	<0.001	<0.001	0.001	0.806
	EPS score	30 (60.0)	35 (70.0)	37 (74.0)	40 (80.0)	0.129	0.296	0.139	0.032	0.656	0.251	0.477
8-week questionnaire survey	FD score	0 (0.0)	1 (2.0)	0 (0.0)	0 (0.0)	0.201	-	-	-	-	-	-
	PDS score	0 (0.0)	1 (2.0)	0 (0.0)	0 (0.0)	0.347	-	-	-	-	-	-
	EPS score	2 (4.0)	1 (2.0)	0 (0.0)	0 (0.0)	0.186	-	-	-	-	-	-

Note: ^{a)} No symptoms: Patients who had symptom resolution (no symptoms) after 4-week treatment, 8-week treatment, 2-week follow-up, or 8-week questionnaire survey.

^{a)}FD score: the composite functional dyspepsia score calculated as the mean of postprandial fullness, early satiety, epigastric pain, and epigastric burning scores. ^{b)}PDS score: the postprandial distress syndrome score is calculated as the mean of postprandial fullness score and early satiety score. ^{c)}EPS score: the epigastric pain syndrome score is calculated as the mean of epigastric pain score and epigastric burning score.

BL-99, *Bifidobacterium animalis* subsp. *Lactis* BL-99.

Patients in the placebo, positive_control, BL-99_low, and BL-99_high groups were administered with maltodextrin (2 g/day), rabeprazole (10 mg/ day), low-dose BL-99 (1×10^{10} CFU/day), and high-dose BL-99 (5×10^{10} CFU/day), respectively.

Question 3: I'm still not convinced the gastrin data fit with colonic microbiome alterations as this study implies because there is no clear cut mechanism. I note the new preliminary animal work but the conclusions this provides "proof" needs toning down. Please add in more details of these experiments: details of the rats (SD means?), how many etc.

Response: Thank you very much for raising this point. The relationship between gastrin and gut microbiota has been clarified in a previous study, which showed that increased production of acetate by an altered gut microbiota led to activation of the parasympathetic nervous system which in turn promoted plasma gastrin concentrations¹. As suggested, we have softened related statements in the revised manuscript to make our conclusions more rigorous. The details are as follows.

1) Original statement: 'Further metagenomic and metabolomics revealed that BL-99 promoted the accumulation of SCFA-producing microbiota and the increase of SCFA levels in stool and serum, thereby regulating the release of gastrin.'

Revised statement: 'Further metagenomic and metabolomics revealed that BL-99 promoted the accumulation of SCFA-producing microbiota and the increase of SCFA levels in stool and serum, which may account for the increase of serum gastrin level.' (Page 1, lines 16-17)

2) Original statement: 'SCFA infusion experiments directly proved that metabolites of gut microbiota can affect serum gastrin level, providing solid evidence that the lower gastrointestinal (fecal) microbiota affects FD.'

Revised statement: 'SCFA infusion experiments demonstrated that metabolites of gut microbiota can affect serum gastrin level in mice, suggesting that the observed increase of serum gastrin in BL-99_high group could be related to the accumulation of SCFA-promoting gut microbiota.' (Page 10, lines 293-296)

And we have added more details of the animal experiments in the Results section (Pages 6-7, lines 172-185) and **Supplementary methods**. The details are shown as follows:

Pages 6-7, lines 172-185: 'Acetate and butyrate were infused into the carotid artery of SD rats for 45 min to elucidate the effect of SCFA on serum gastrin. The serum gastrin levels were determined at 0, 15, 30, and 45 min respectively, and the detailed animal experiment method was shown in the **Supplementary methods**. SCFA infusion results showed that 8 $\mu\text{mol}/(\text{kg}\cdot\text{min})$ acetate, 20 $\mu\text{mol}/(\text{kg}\cdot\text{min})$ acetate, and 1 $\mu\text{mol}/(\text{kg}\cdot\text{min})$ butyrate increased serum gastrin levels after perfusion. Moreover, serum gastrin in the 20 $\mu\text{mol}/(\text{kg}\cdot\text{min})$ acetate group ($365.68 \pm 25.03 \text{ pg/mL}$) was significantly higher than that in the 8 $\mu\text{mol}/(\text{kg}\cdot\text{min})$ acetate group ($270.27 \pm 23.00 \text{ pg/mL}$, $P < 0.001$) and the 2 $\mu\text{mol}/(\text{kg}\cdot\text{min})$ acetate group ($201.86 \pm 35.15 \text{ pg/mL}$, $P = 0.001$) after acetate infusion for 45 min. Similar results were found in the butyrate infusion experiment, which showed that serum gastrin in the 1 $\mu\text{mol}/(\text{kg}\cdot\text{min})$ acetate group ($310.10 \pm 25.94 \text{ pg/mL}$) was significantly higher than that in the 0.5 $\mu\text{mol}/(\text{kg}\cdot\text{min})$ acetate group ($210.50 \pm 26.68 \text{ pg/mL}$, $P < 0.001$) and the 0.1 $\mu\text{mol}/(\text{kg}\cdot\text{min})$ acetate group ($191.15 \pm 29.75 \text{ pg/mL}$, $P < 0.001$) (**Supplementary Fig. 5**). This suggests that acetate and butyrate maybe affect the serum gastrin level of FD patients.'

Supplementary methods

SCFA infusion experiment

Ethics statement

All the experimental procedures were approved by the Ethics Committee of Beijing Laboratory Animal Research Center (approval No. BLARC-LAWER-202306006), and were also in

accordance with the Guidelines for the Care and Use of Laboratory Animals published by the US National Institutes of Health.

Animals

Thirty-six normal male Sprague-Dawley rats (8-9 weeks of age, 350-390 g) were ordered from Charles River Laboratories (CRL, Beijing, China). All animals were housed in the SPF animal housing facility with a 12-h on/12-h off light cycle, 20-26 °C ambient temperature, 40-70% humidity, and free access to food (batch No. 0515SH05190325C) and water. And all animals were allowed to acclimate for 7 d, fasted for 12 h (overnight), and randomly divided into 6 groups (n=6) prior to SCFA infusion experiments.

Intracarotid SCFA infusion experiments

The intracarotid SCFA infusion experiments were performed following previously described protocol^[11], with some modification. Rats were general anesthetized with isoflurane (2% induction, 2% maintenance in 70% N₂ and 30% O₂). A femoral artery catheter was used to monitor arterial blood pressure. Rectal temperature was monitored and maintained at 37 °C via a servo-controlled heating pad. PE50 tubing was inserted retrogradely into the external carotid artery and advanced into the carotid bifurcation. And then 2, 8, 20 μmol/(kg-min) acetate and 0.1, 0.5, 1 μmol/(kg-min) butyrate were infused immediately for 45 min, and the animals were euthanized. Blood samples (200 μL) were collected during the infusion period (0, 15, 30, and 45 min). Serum sample was supernatants obtain from blood samples after centrifugation at 3000 r.p.m. for 15 min. And serum gastrin was measured using mlbio ELISA kit (Shanghai Enzyme-linked Biotechnology Co., Ltd, China)

Statistical analysis

Serum gastrin levels were described as mean (M) ± standard deviation (SD). And significant differences among different groups were evaluated by one-way analysis of variance (ANOVA) with least significant difference (LSD) analysis.

Reference

1. Perry, R. J. et al. Acetate mediates a microbiome-brain-β-cell axis to promote metabolic syndrome. *Nature (London)*. **534**, 213-217 (2016)

Question 4: The Maastricht consensus refers to *H. pylori* positive patients in general. As this was a Hp negative cohort the relevance of referring to this consensus in supporting testing gastrin and pepsinogen testing seems irrelevant.

Response: Thanks for your reminding. We have revised the description in the Discussion section (Page 8, lines 225-227):

‘Paloheimo, L. *et al.* reported that pepsinogen and gastrin as biomarkers can provide information about the structure and function of gastric mucosa, which can be used to assist in the examination of dyspeptic symptoms¹.’

Reference

1. Paloheimo, L. et al. Serological biomarker test (GastroPanel®) in the diagnosis of functional gastric disorders, *Helicobacter pylori* and atrophic gastritis in patients examined for dyspeptic symptoms. *Anticancer Res.* **41**, 811-819 (2021).

Reviewer #3 (Remarks to the Author):

The authors have taken care to address all comments made by reviewers, and this has led to a much improved manuscript.

Response: Thank you very much for your overall positive comments.

Reviewer #4 (Remarks to the Author):

Summary: The current treatments for functional dyspepsia (FD) are not consistently effective. This study was conducted where 200 FD patients were given either a placebo, a known drug (rabeprazole), or different doses of a probiotic called *Bifidobacterium animalis* subsp. *Lactis* BL-99 for 8 weeks. The main measure was the clinical response rate (CRR) of FD, specifically if there was a decrease in FD score by more than 0.5. Results showed that the high dose of BL-99 had a 95.6% CRR, which was notably better than the placebo, low dose BL-99, and the known drug. This positive effect from BL-99 lasted for 2 weeks post-treatment but vanished by the 8th week after treatment. Further analysis indicated that BL-99 boosts the presence of SCFA-producing bacteria and raises SCFA levels in both stool and blood, which in turn affects the release of a hormone called gastrin. This suggests that BL-99 has potential as a sustainable treatment for FD relief.

The study is of interest but the results are not particularly novel and shows modest effects. Moreover, the study provides no clear convincing arguments for the mechanisms proposed to mediate the beneficial effect. Both the appearance of SCFA and the modulation of gastrin level had been reported before.

Response: We thank the reviewer for this valuable comment. Although the efficacy of probiotics on FD were reported in some studies, evidence about the prolonged effect of probiotics after treatment is lacking. Moreover, the appearance of SCFA and the modulation of gastrin levels had been reported in *Helicobacter pylori*-infected individuals¹, few studies were conducted among FD patients without *Helicobacter pylori*-infection. In this study among FD patients without *Helicobacter pylori*-infection, we observed that the clinical response rate in FD score of the BL-99_high group was significantly higher than that of placebo and positive_control groups after 8-week treatment, and this effect persisted until 2 weeks after treatment. Based on the significant efficacy found in this study, metagenomic and metabolomics methods were further used to explore the possible mechanism and proposed a possible pathway (i.e. microbiota-SCFA-gastrin pathway) by which BL-99 improves FD. However, as the reviewer pointed out, the exact mechanism needs to be confirmed by further research. Our study provides a potential therapeutic avenue for alleviating symptoms in FD patients without *Helicobacter pylori*-infection; however, further researches are warranted to confirm the underlying mechanism.

Reference

1. Myllyluoma, E. et al. Probiotic intervention decreases serum gastrin-17 in *Helicobacter pylori* infection. *Dig. Liver Dis.* **39**, 516-523 (2007).

In detail:

Question 1: The confounding effects of increased body weights was not assessed. Hence the population studied is not really representative, same applies to gender distribution.

Response: We thank the reviewer for pointing out this issue.

Regarding the representation of gender distribution, studies have confirmed that the prevalence of FD is higher in women than in men. Sperber, A. D. *et al* collected the prevalence of GI diseases in 33 countries/regions, and the results showed that the average prevalence of FD was higher in women than in men¹. Ford, A. C., *et al* suggested that female sex is an independent risk factor for FD². Both the United European Gastroenterology (UEG)/ European Society for Neuro

gastroenterology and Motility (ESNM) consensus and Chinese expert consensus on functional dyspepsia pointed out that the prevalence of FD in women was higher than that in men^{3,4}. In our study, the percentage of female subjects was higher than that of male subjects, which is consistent with previous studies. Further subgroup analysis by gender was also conducted in our analysis to provide more detail on the effects of gender.

There is no clear evidence for a relationship between body weight or BMI and FD. To elucidate this representativeness, we also performed a subgroup analysis based on BMI for the primary outcome.

Subgroup analyses by gender and BMI were shown in the revised manuscript as follows (Page 4, lines 98-102):

‘The results of CRR were analyzed separately in men and women, in patients with BMI < 24 and BMI \geq 24 kg/m² based on ITT as well (**Supplementary Table 2, Table 3, Table 4, and Table 5**). Overall, the results of the primary outcome for women, BMI < 24 and BMI \geq 24 kg/m² were consistent with those for the total population. Due to the relatively small proportion of male participants (25.5%), no significant effects were found in men.’

Reference

1. Sperber, A. D. et al. Worldwide prevalence and burden of functional gastrointestinal disorders, results of Rome foundation global study. *Gastroenterology*. **160**, 99-114 (2021).
2. Ford, A. C. et al. Functional dyspepsia. *The Lancet*. **396**, 1689-1702 (2020).
3. Wauters, L. et al. United European Gastroenterology (UEG) and European Society for Neuro gastroenterology and Motility (ESNM) consensus on functional dyspepsia. *United European Gastroenterol. J.* **9**, 307-331 (2021).
4. Study Group of Gastrointestinal Motility et al. Chinese consensus on the management of functional dyspepsia (2015, Shanghai). *Chin J Dig.* **36**, 217-229 (2016).

Supplementary Table 2 The clinical response rate for men participants based on intention-to-treat (ITT) set

Clinical response rate ^{a)} , No. (%)		Placebo (n=13)	Positive_control (n=12)	BL-99_low (n=13)	BL-99_high (n=13)	P _{overall}	P					
							Positive_control vs	BL-99_low vs	BL-99_high vs	BL-99_low vs	BL-99_high vs	BL-99_high vs
							Placebo	Placebo	Placebo	Positive_control	Positive_control	BL-99_low
4-week treatment	FD score ^{a)}	6 (46.2)	9 (75.0)	9 (69.2)	11 (84.6)	0.185	-	-	-	-	-	-
	PDS score ^{b)}	7 (53.8)	8 (66.7)	11 (84.6)	13 (100.0)	0.032	0.515	0.102	0.998	0.303	0.999	0.999
	EPS score ^{c)}	9 (69.2)	8 (66.7)	9 (69.2)	11 (84.6)	0.729	-	-	-	-	-	-
8-week treatment	FD score	10 (76.9)	9 (75.0)	9 (69.2)	13 (100.0)	0.211	-	-	-	-	-	-
	PDS score	11 (84.6)	10 (83.3)	10 (76.9)	13 (100.0)	0.370	-	-	-	-	-	-
	EPS score	9 (69.2)	8 (66.7)	8 (61.5)	11 (84.6)	0.605	-	-	-	-	-	-
2-week follow-up	FD score	8 (61.5)	9 (75.0)	9 (69.2)	13 (100.0)	0.111	-	-	-	-	-	-
	PDS score	9 (69.2)	9 (75.0)	10 (76.9)	13 (100.0)	0.211	-	-	-	-	-	-
	EPS score	9 (69.2)	10 (83.3)	9 (69.2)	13 (100.0)	0.152	-	-	-	-	-	-
8-week questionnaire survey	FD score	5 (38.5)	6 (50.0)	1 (7.7)	5 (38.5)	0.130	-	-	-	-	-	-
	PDS score	5 (38.5)	5 (41.7)	3 (23.1)	6 (46.2)	0.644	-	-	-	-	-	-
	EPS score	3 (23.1)	4 (33.3)	1 (7.7)	6 (46.2)	0.160	-	-	-	-	-	-

Note: ^{a)}Clinical response rate was defined as the proportion of participants with a score (i.e., FD score, PDS score, and EPS score) decrease >0.5.

^{a)}FD score: the composite functional dyspepsia score is calculated as the mean of postprandial fullness, early satiety, epigastric pain, and epigastric burning scores. ^{b)}PDS score: the postprandial distress syndrome score is calculated as the mean of postprandial fullness score and early satiety score. ^{c)}EPS score: the epigastric pain syndrome score is calculated as the mean of epigastric pain score and epigastric burning score.

BL-99, *Bifidobacterium animalis* subsp. *Lactis* BL-99.

Patients in the placebo, positive_control, BL-99_low, and BL-99_high groups were administered with maltodextrin (2 g/day), rabeprazole (10 mg/ day), low-dose BL-99 (1×10^{10} CFU/day), and high-dose BL-99 (5×10^{10} CFU/day), respectively.

Supplementary Table 3 The clinical response rate for women participants based on intention-to-treat (ITT) set

Clinical response rate ^{a)} , No. (%)		Placebo (n=37)	Positive_control (n=38)	BL-99_low (n=37)	BL-99_high (n=37)	P overall	P					
							Positive_control vs	BL-99_low vs	BL-99_high vs	BL-99_low vs	BL-99_high vs	BL-99_high vs
							Placebo	Placebo	Placebo	Positive_control	Positive_control	BL-99_low
4-week treatment	FD score ^{a)}	23 (62.2)	19 (50.0)	19 (51.4)	27 (73.0)	0.151	-	-	-	-	-	-
	PDS score ^{b)}	24 (64.9)	24 (63.2)	26 (70.3)	30 (81.1)	0.326	-	-	-	-	-	-
	EPS score ^{c)}	15 (40.5)	15 (39.5)	16 (43.2)	24 (64.9)	0.092	-	-	-	-	-	-
8-week treatment	FD score	19 (51.4)	26 (68.4)	28 (75.7)	32 (86.5)	0.009	0.134	0.032	0.002	0.485	0.068	0.241
	PDS score	23 (62.2)	29 (76.3)	30 (81.1)	31 (83.8)	0.132	-	-	-	-	-	-
	EPS score	15 (40.5)	19 (50.0)	23 (62.2)	26 (70.3)	0.051	-	-	-	-	-	-
2-week follow-up	FD score	23 (62.2)	24 (63.2)	29 (78.4)	29 (78.4)	0.219	-	-	-	-	-	-
	PDS score	27 (73.0)	29 (76.3)	34 (91.9)	31 (83.8)	0.160	-	-	-	-	-	-
	EPS score	17 (45.9)	20 (52.6)	23 (62.2)	25 (67.6)	0.238	-	-	-	-	-	-
8-week questionnaire survey	FD score	5 (13.5)	2 (5.3)	8 (21.6)	11 (29.7)	0.035	0.235	0.363	0.097	0.053	0.012	0.426
	PDS score	7 (18.9)	11 (28.9)	15 (40.5)	13 (35.1)	0.214	-	-	-	-	-	-
	EPS score	2 (5.4)	5 (13.2)	5 (13.5)	9 (24.3)	0.136	-	-	-	-	-	-

Note: ^{a)}Clinical response rate was defined as the proportion of participants with a score (i.e., FD score, PDS score, and EPS score) decrease >0.5.

^{a)}FD score: the composite functional dyspepsia score is calculated as the mean of postprandial fullness, early satiety, epigastric pain, and epigastric burning scores. ^{b)}PDS score: the postprandial distress syndrome score is calculated as the mean of postprandial fullness score and early satiety score. ^{c)}EPS score: the epigastric pain syndrome score is calculated as the mean of epigastric pain score and epigastric burning score.

BL-99, *Bifidobacterium animalis* subsp. *Lactis* BL-99.

Patients in the placebo, positive_control, BL-99_low, and BL-99_high groups were administered with maltodextrin (2 g/day), rabeprazole (10 mg/ day), low-dose BL-99 (1×10^{10} CFU/day), and high-dose BL-99 (5×10^{10} CFU/day), respectively.

Supplementary Table 4 The clinical response rate in participants with BMI < 24 kg/m² based on intention-to-treat (ITT) set

Clinical response rate ^{a)} , No. (%)		Placebo (n=19)	Positive_control (n=22)	BL-99_low (n=20)	BL-99_high (n=19)	P _{overall}	P					
							Positive_control vs	BL-99_low vs	BL-99_high vs	BL-99_low vs	BL-99_high vs	BL-99_high vs
							Placebo	Placebo	Placebo	Positive_control	Positive_control	BL-99_low
4-week treatment	FD score ^{a)}	11 (57.9)	11 (50.0)	11 (55.0)	14 (73.7)	0.464	-	-	-	-	-	-
	PDS score ^{b)}	12 (63.2)	13 (59.1)	16 (80.0)	17 (89.5)	0.106	-	-	-	-	-	-
	EPS score ^{c)}	8 (42.1)	8 (36.4)	8 (40.0)	14 (73.7)	0.072	-	-	-	-	-	-
8-week treatment	FD score	8 (42.1)	16 (72.7)	15 (75.0)	16 (84.2)	0.030	0.051	0.041	0.011	0.867	0.381	0.480
	PDS score	9 (47.4)	17 (77.3)	16 (80.0)	16 (84.2)	0.043	0.053	0.039	0.022	0.830	0.578	0.732
	EPS score	7 (36.8)	11 (50.0)	13 (65.0)	14 (73.7)	0.101	-	-	-	-	-	-
2-week follow-up	FD score	11 (57.9)	12 (54.5)	16 (80.0)	16 (84.2)	0.094	-	-	-	-	-	-
	PDS score	15 (78.9)	16 (72.7)	19 (95.0)	18 (94.7)	0.106	-	-	-	-	-	-
	EPS score	7 (36.8)	10 (45.5)	13 (65.0)	14 (73.7)	0.078	-	-	-	-	-	-
8-week questionnaire survey	FD score	2 (10.5)	2 (9.1)	4 (20.0)	3 (15.8)	0.730	-	-	-	-	-	-
	PDS score	2 (10.5)	9 (40.9)	7 (35.0)	5 (26.3)	0.164	-	-	-	-	-	-
	EPS score	0 (0.0)	2 (9.1)	2 (10.0)	2 (10.5)	0.562	-	-	-	-	-	-

Note: ^{a)}Clinical response rate was defined as the proportion of participants with a score (i.e., FD score, PDS score, and EPS score) decrease >0.5.

^{a)}FD score: the composite functional dyspepsia score is calculated as the mean of postprandial fullness, early satiety, epigastric pain, and epigastric burning scores. ^{b)}PDS score: the postprandial distress syndrome score is calculated as the mean of postprandial fullness score and early satiety score. ^{c)}EPS score: the epigastric pain syndrome score is calculated as the mean of epigastric pain score and epigastric burning score.

BL-99, *Bifidobacterium animalis* subsp. *Lactis* BL-99.

Patients in the placebo, positive_control, BL-99_low, and BL-99_high groups were administered with maltodextrin (2 g/day), rabeprazole (10 mg/ day), low-dose BL-99 (1×10^{10} CFU/day), and high-dose BL-99 (5×10^{10} CFU/day), respectively.

Supplementary Table 5 The clinical response rate in participants with BMI ≥ 24 kg/m² based on intention-to-treat (ITT) set

Clinical response rate ^{a)} , No. (%)		Placebo (n=31)	Positive_control (n=28)	BL-99_low (n=30)	BL-99_high (n=31)	P _{overall}	P					
							Positive_control vs	BL-99_low vs	BL-99_high vs	BL-99_low vs	BL-99_high vs	BL-99_high vs
							Placebo	Placebo	Placebo	Positive_control	Positive_control	BL-99_low
4-week treatment	FD score ^{a)}	18 (58.1)	17 (60.7)	17 (56.7)	24 (77.4)	0.299	-	-	-	-	-	-
	PDS score ^{b)}	19 (61.3)	19 (67.9)	21 (70.0)	26 (83.9)	0.256	-	-	-	-	-	-
	EPS score ^{c)}	16 (51.6)	15 (53.6)	17 (56.7)	21 (67.7)	0.582	-	-	-	-	-	-
8-week treatment	FD score	21 (67.7)	19 (67.9)	22 (73.3)	29 (93.5)	0.058	0.992	0.633	0.019	0.647	0.021	0.048
	PDS score	25 (80.6)	22 (78.6)	24 (80.0)	28 (90.3)	0.611	-	-	-	-	-	-
	EPS score	17 (54.8)	16 (57.1)	18 (60.0)	23 (74.2)	0.400	-	-	-	-	-	-
2-week follow-up	FD score	20 (64.5)	21 (75.0)	22 (73.3)	26 (83.9)	0.384	-	-	-	-	-	-
	PDS score	21 (67.7)	22 (78.6)	25 (83.3)	26 (83.9)	0.384	-	-	-	-	-	-
	EPS score	19 (61.3)	20 (71.4)	19 (63.3)	24 (77.4)	0.500	-	-	-	-	-	-
8-week questionnaire survey	FD score	8 (25.8)	6 (21.4)	5 (16.7)	13 (41.9)	0.131	-	-	-	-	-	-
	PDS score	10 (32.3)	7 (25.0)	11 (36.7)	14 (45.2)	0.427	-	-	-	-	-	-
	EPS score	5 (16.1)	7 (25.0)	4 (13.3)	13 (41.9)	0.039	0.401	0.759	0.030	0.264	0.174	0.017

Note: ^{a)}Clinical response rate was defined as the proportion of participants with a score (i.e., FD score, PDS score, and EPS score) decrease >0.5 .

^{a)}FD score: the composite functional dyspepsia score is calculated as the mean of postprandial fullness, early satiety, epigastric pain, and epigastric burning scores. ^{b)}PDS score: the postprandial distress syndrome score is calculated as the mean of postprandial fullness score and early satiety score. ^{c)}EPS score: the epigastric pain syndrome score is calculated as the mean of epigastric pain score and epigastric burning score.

BL-99, *Bifidobacterium animalis* subsp. *Lactis* BL-99.

Patients in the placebo, positive_control, BL-99_low, and BL-99_high groups were administered with maltodextrin (2 g/day), rabeprazole (10 mg/day), low-dose BL-99 (1×10^{10} CFU/day), and high-dose BL-99 (5×10^{10} CFU/day), respectively.

Question 2: No objective endpoint was applied to measure the effects observed.

Response: We understand the reviewer's concern. Functional dyspepsia (FD) is a common gastrointestinal disorder characterized by dyspeptic symptoms¹. There were no specific objective indicators to evaluate FD. Symptom questionnaires including FD score, PDS score and EPS score are commonly used in clinical practice and related studies^{2,3}

Reference

1. Sayuk, G. S. et al. Functional dyspepsia: Diagnostic and therapeutic approaches. *Drugs (New York, N.Y.)*. **80**, 1319-1336 (2020).
2. Wauters, L. et al. Efficacy and safety of spore-forming probiotics in the treatment of functional dyspepsia: a pilot randomised, double-blind, placebo-controlled trial. *Lancet Gastroenterol. Hepatol.* **6**, 784-792 (2021).
3. Wauters, L. et al. United European Gastroenterology (UEG) and European Society for Neuro gastroenterology and Motility (ESNM) consensus on functional dyspepsia. *United European Gastroenterol. J.* **9**, 307-331 (2021).

Question 3: The effect would have to be corroborated by using a different bacterial strain in order to proof that the effect is strain specific.

Response: Thank the reviewer for this valuable comment. We fully agree with you that the effects of probiotics on human health are strain-specific, which is why the specific health benefits of certain probiotics often need to be proved by dedicated and rigorous clinical studies. The main purpose of this study was to investigate the prolonged efficacy of *Bifidobacterium animalis* subsp. *lactis* BL-99 in the treatment of FD. We designed a rigorous randomized, parallel-group, positive-drug, and placebo-controlled clinical trial to explore the efficacy of BL-99 on FD. Sufficient clinical data including symptom score, serology, and multi-omics analysis showed that BL-99 may alleviate FD symptoms by regulating gut microbiota, SCFA, and gastrin. Thus, the main purpose of this study has been achieved based on the existing design, research process, and results. For other strains, additional clinical studies are needed to explore their effectiveness on FD.

Question 4: The differential expression of SCFA during probiotic treatment is rather expected and likely not strain specific.

Response: Thank you very much for raising this point. Although the differential expression of SCFAs during probiotic treatment is rather expected, strain-specific issues of SCFAs production need to be elucidated. Firstly, probiotic strains could secrete different types of SCFAs. For example, *Bacillus coagulans* and *Saccharomyces boulardii* CNCM I-745 were able to produce the three kinds of SCFAs including acetate, propionate, and butyrate¹, while *Bifidobacterium breve* UCC2003 only secreted acetate². Secondly, probiotic strains could change the composition of gut microbiota and further change the types and amounts of SCFA produced by gut microbiota³. Such as *Lactobacillus plantarum* Dad-13 consumption increased the relative abundance of *Faecalibacterium* and *Bifidobacterium*, which was associated with the increment in propionate and butyrate⁴. Moreover, *Bifidobacterium bifidum* CCFM16 significantly enhanced the relative abundance of *Clostridia* and then increased the acetate and butyrate concentrations⁵. Thirdly, the same probiotic strain has different effects on intestinal SCFA production in different target populations. For example, *Bifidobacterium animalis* subsp. *lactis* BB-12 (BB-12) supplementation

was associated with a more rapid return to baseline acetate levels in antibiotic-induced acetate reduction individuals⁶. While, BB-12 did not significantly alter the fecal SCFA concentration in healthy young adults⁷. What's more, different kinds of SCFA have different physiological functions. For example, only supplementation with butyrate reduced arthritis, but not acetate and propionate⁸. Therefore, although the production of SCFA in probiotic interventions is expected, the types and amounts of SCFA in the intestine of different intervention strains and different target populations still need to be explored.

Reference

1. Calvigioni, M. et al. HPLC-MS-MS quantification of short-chain fatty acids actively secreted by probiotic strains. *Front. Microbiol.* **14**, 1124144 (2023).
2. Ruiz-Aceituno, L. et al. Metabolism of biosynthetic oligosaccharides by human-derived *Bifidobacterium breve* UCC2003 and *Bifidobacterium longum* NCIMB 8809. *Int. J. Food Microbiol.* **316**, 108476 (2020).
3. Markowiak-Kopeć, et al. The effect of probiotics on the production of short-chain fatty acids by human intestinal microbiome. *Nutrients.* **12**, 1107 (2020).
4. Kamil, R. Z. et al. Gut microbiota modulation of moderate undernutrition in infants through gummy *Lactobacillus plantarum* Dad-13 consumption: A randomized double-blind controlled trial. *Nutrients.* **14**, (2022).
5. Wang, L. et al. A randomised, double-blind, placebo-controlled trial of *Bifidobacterium bifidum* CCFM16 for manipulation of the gut microbiota and relief from chronic constipation. *Food Funct.* **13**, 1628-1640 (2022).
6. Merenstein, D. et al. *Bifidobacterium animalis* subsp. lactis BB-12 protects against antibiotic-induced functional and compositional changes in human fecal microbiome. *Nutrients.* **13**, (2021).
7. Ba, Z. et al. Matrix effects on the delivery efficacy of *Bifidobacterium animalis* subsp. lactis BB-12 on fecal microbiota, gut transit time, and short-chain fatty acids in healthy young adults. *Mosphere.* **6**, e8421 (2021).
8. Rosser, E. C. et al. Microbiota-derived metabolites suppress arthritis by amplifying aryl-hydrocarbon receptor activation in regulatory B cells. *Cell Metab.* **31**, 837-851 (2020).

Question 5: The influence of SCFA on gastrin levels in humans and rodents is no proof of efficacy of this probiotic treatment.

Response: Thank you for your question. In fact, the efficacy of probiotics in the treatment of FD was evaluated mainly by symptom score¹. In our study, the main purpose was to investigate the prolonged efficacy and the possible reasons for BL-99 in the treatment of FD. So the FD symptom score was chosen as the primary outcome, and SCFA and serum gastrin were mainly used to explore the possible mechanism for BL-99 relieving FD. However, the exact mechanism needs to be confirmed by further studies.

Reference

1. Wauters, L. et al. Efficacy and safety of spore-forming probiotics in the treatment of functional dyspepsia: a pilot randomised, double-blind, placebo-controlled trial. *Lancet Gastroenterol. Hepatol.* **6**, 784-792 (2021).

Question 6: There is no data on the evaluation of food frequency questionnaires in association with the changes in microbiota.

Response: We thank the reviewer for this valuable comment. We fully agree with you that dietary patterns are correlated with gut microbiome composition¹. We did not collect dietary

information during this study. However, participants in this study were asked to maintain their habitual eating habits to reduce the impact of dietary changes on their microbiota. (Page 11, lines 339-341). We also believe that the evaluation of dietary data during the study could elucidate the effects of diet on microbiota in addition to the BL99 treatment, which would support the effects of BL99 on microbiota in this study. So we have added a discussion of this limitation in the revised manuscript (Page 9, lines 273-275).

‘Fourthly, although FD participants were required to maintain their dietary habits during the study, no dietary survey was conducted to assess the effects of diet on gut microbiota composition.’

Reference

1. Bolte, L. A. et al. Long-term dietary patterns are associated with pro-inflammatory and anti-inflammatory features of the gut microbiome. *Gut*. **70**, 1287-1298 (2021).

Question 7: Were the methodologies applied all from the very same fecal isolate or were different portions of the stool samples used for analysis. If the entire metabolites are not taken and worked up for the entire of DNA, metabolites etc, an enormous bias can be expected.

Response: Thank you for your question. Our fecal samples were homogenized by Bertin Precellys Evolution sample homogenizer (Bertin Technologies SAS, France) referring to the previous study with some modification. And we have supplemented the detailed information in the Method section (Pages 12-13, lines 378-381):

‘More importantly, fecal samples were homogenized by Bertin Precellys Evolution sample homogenizer (Bertin Technologies SAS, France) referring to the previous study^{1,2}, and then the homogenized fecal samples were randomly weighed for further index detection.’

Reference

1. Moosmang, S. et al. Metabolomic analysis-Addressing NMR and LC-MS related problems in human feces sample preparation. *Clin. Chim. Acta*. **489**, 169-176 (2019).
2. Paramsothy, S. et al. Specific bacteria and metabolites associated with response to fecal microbiota transplantation in patients with ulcerative colitis. *Gastroenterology (New York, N.Y. 1943)*. **156**, 1440-1454 (2019).

REVIEWERS' COMMENTS

Reviewer #1 (Remarks to the Author):

The paper is improved with the revisions.

In discussion the authors state gastrin and pepsinogen testing is a biomarker of dyspepsia. This is misleading. These alterations are confounded by H. pylori infection (current and past) and the presence or absence of atrophic gastritis. Further, it is well established there is no association between FD and gastric acid secretion levels (which are similar to controls). I'd recommend this paragraph be revised.

Point-by-point response to the reviewers' comments

Reviewer #1 (Remarks to the Author):

The paper is improved with the revisions.

Question 1: In discussion the authors state gastrin and pepsinogen testing is a biomarker of dyspepsia. This is misleading. These alterations are confounded by *H. pylori* infection (current and past) and the presence or absence of atrophic gastritis. Further, it is well established there is no association between FD and gastric acid secretion levels (which are similar to controls). I'd recommend this paragraph be revised.

Response: Thanks for your suggestion. The insights you have provided are remarkably profound and enlightening to our research. In accordance with your invaluable suggestions, we have meticulously revised the discussion in this paragraph to enhance the accuracy of our expression. (Page 9, lines 253-266). The details are as follows, with the modified content highlighted in red.

(Page 9, lines 253-266): 'Studies have confirmed that serum pepsinogen and gastrin levels in FD patients are different from those in healthy persons and are associated with various symptoms of FD¹⁻³. For instance, Tahara, T. et al.² discovered that serum PGII levels were significantly elevated and the PGI/II ratio was significantly reduced in both *H. pylori* positive and negative FD patients compared to healthy controls. Furthermore, Igarashi *et al.* found that *Lactobacillus gasseri* OLL2716 increased serum PGI levels in FD patients and other functional upper gastrointestinal disorders patients⁴. Additionally, G17, an important gastrointestinal hormone, has also been reported to be potentially related to FD, and it was revealed that acupuncture, a type of traditional Chinese medical therapy, improved FD symptoms and increased serum G17 levels⁵. Therefore, PGI, PGII, and G17 were determined in this study. Our results showed that BL-99 had no significant effect on serum PG level after 8-week treatment, but increased serum G17 levels in FD participants compared with placebo and positive_control groups. So our results indicated that BL-99 treatment could regulate the changes in gastrin associated with FD.'

Reference

1. Crafa, P., et al. Functional dyspepsia. *Acta Biomed.* **91**, e2020069 (2020).
2. Tahara, T., et al. Examination of serum pepsinogen in functional dyspepsia. *Hepatogastroenterology.* **59**, 2516-2522 (2012).
3. Paloheimo, L., et al. Serological biomarker test (GastroPanel®) in the diagnosis of functional gastric disorders, *Helicobacter pylori* and atrophic gastritis in patients examined for dyspeptic symptoms. *Anticancer Res.* **41**, 811-819 (2021).
4. Igarashi, M., et al. Correlation between the serum pepsinogen I level and the symptom degree in proton pump inhibitor-Users administered with a probiotic. *Pharmaceuticals.* **7**, 754-764 (2014).
5. Kwon, C. Y., et al. Acupuncture as an add-on treatment for functional dyspepsia: A systematic review and meta-analysis. *Front. Med.* **8**, 682783 (2021).